# PIECEWISE POLYNOMIAL REGRESSION OF TAME FUNCTIONS VIA INTEGER PROGRAMMING

## ABSTRACT

Tame functions are a class of nonsmooth, nonconvex functions that appear in a wide range of applications: in training deep neural networks with all common activations, as value functions of mixed-integer programs, or as wave functions of small molecules. We consider approximating tame functions with piecewise polynomial functions. We present a theoretical bound on the approximation quality of a tame function by a piecewise polynomial function. We also present mixed-integer programming formulations of piecewise polynomial regression and demonstrate promising computational results.

## 1 INTRODUCTION

In a wide range of applications, one encounters nonsmooth and nonconvex functions that are *tame*, short for *definable in o-minimal structures*. Crucially, the domain of each tame function can be decomposed into a finite number of regions on which the function behaves smoothly; see Fig. 1 (left pane). Usual classes of nonsmooth nonconvex functions, such as Lipschitz continuous and weakly convex, do not generally benefit from such properties. Tame functions appear in a broad range of useful and difficult, i.e., nonsmooth and nonconvex, applications. Prominent examples are all common deep learning architectures (Davis et al., 2020; Bolte & Pauwels, 2020; Bareilles et al., 2025), and empirical risk minimization frameworks (Iutzeler & Malick, 2020). Tame functions also appear, e.g., in mixed-integer optimization, with the value function and the solution to the so-called subadditive dual (Aspman et al., 2024); in quantum information theory, with approximations of the matrix exponential for a k-local Hamiltonian (Bondar et al., 2022; Aravanis et al., 2022); and in quantum chemistry, with functions describing the electronic structure of molecules. The tame assumption has been key in recent advances in learning theory, with notably the first convergence proofs of Stochastic Gradient Descent (Davis et al., 2020; 2025; Bianchi et al., 2023), or theory of automatic differentiation (Bolte & Pauwels, 2020). The tameness property of a function is, among other things, stable under composition.

This paper is concerned with building approximation of nonsmooth nonconvex functions, a topic which has received attention recently (Huchette & Vielma, 2023; Chatterjee & Goswami, 2021), building on older results (Donoho, 1997), under the tame assumption. Formally, given a complex function $f$, we seek a function $p$ in a set of simple functions $\mathcal{P}$ that minimizes the distance from $p$ to $f$ over a domain $A$:

$$\inf_{p \in \mathcal{P}} \left( \|f - p\|_{\infty, A} = \sup_{x \in A} |f(x) - p(x)| \right).$$

A major challenge is the nonsmoothness of the function to approximate. Classical polynomial approximation theory shows that good ("fast") polynomial approximation of a function is possible if the function has a high degree of regularity. Specifically, if $f$ is a function $m$-times continuously differentiable, the best degree $N$ polynomial incurs an $\mathcal{O}(1/N^m)$ error (Bagby et al., 2002). However, the situation changes dramatically when the function has low regularity, e.g., is continuous but with discontinuous derivatives, as is the case for most settings in learning theory. Indeed, approximating the absolute value — the simplest nonsmooth function — by degree $N$ polynomials over the interval $[-1, 1]$ incurs *exactly* the slow rate $1/N$:

$$\inf_{p \in \mathcal{P}_N} \|p - |\cdot|\|_{\infty, [-1,1]} = \frac{\beta}{N} + o\left(\frac{1}{N}\right),$$

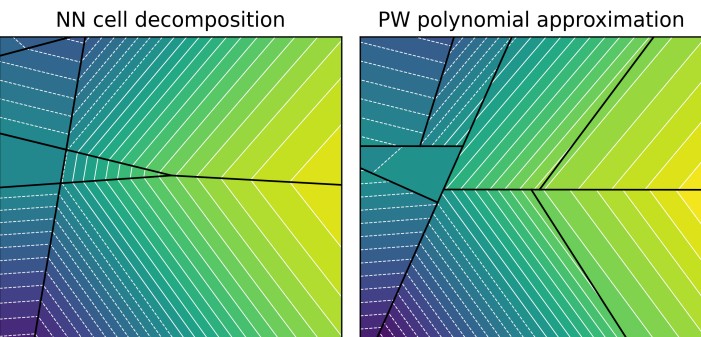

**Figure 1:** Left pane: a generic 2-dimensional function modeled by a neural network, with level lines (white) and nonsmooth points (black). Right pane: piecewise polynomial approximation of the network, obtained from the proposed integer program with a depth 3 regression tree and degree 2 polynomials.

where $\beta \approx 0.28$ (Bernstein, 1914). In sharp contrast, allowing an approximation by *piecewise polynomial* function makes this problem simpler: the absolute value itself is a piecewise polynomial, consisting of two polynomial pieces of degree 1. As a second example, consider the neural network comprised of sigmoid, tanh, and ReLU activations depicted in Fig. 1 (left pane). Observe that the nonsmooth points of the network, denoted by solid black lines, are well-behaved: they delineate a finite number of regions on which the function behaves smoothly, as guaranteed by the tameness property. In this work, we explore the approximation of such functions by a piecewise polynomial function. Inded, we are not aware of any convergent method for finding a piecewise polynomial approximation of such nonlinear networks, or more generally arbitrary tame functions.

**Related work** While simple, these examples illustrate the two fundamental challenges of estimating nonsmooth functions:

  (i) estimating the *stratification* of the function, that is the collection of full-dimensional sets on which the function is smooth; and,
  (ii) estimating the function on each such stratum.

These challenges require input from several branches of mathematics.

For *(i)*, one can consult Model Theory, and, more specifically, o-minimal structures. Let us recall a central theorem there: the domain of any tame function partitions into *finitely* many smooth sets, known as "strata", on which the function can have any desired degree of smoothness (Van den Dries, 1998). A natural approach is then to estimate this stratification of the space. Rannou (1998) reduces this to a quantifier elimination problem and proposes a procedure that takes doubly-exponential time, for semialgebraic functions. We are not aware of any implementation of this approach. Helmer & Nanda (2023a;b); Helmer & Mohr (2024) tested an implementation for real algebraic varieties and, possibly, semialgebraic sets. We propose an algorithm that applies to general tame functions, thus covering the semialgebraic case but also networks using e.g., sigmoid or tanh activations. Boissonnat et al. (2023); Boissonnat & Wintraecken (2022) deal with the related task of building a piecewise linear approximation of a given smooth manifold that is explicitly available, by means of its distance in broad terms. They provide a algorithm that allows to construct a piecewise linear approximation of the smooth manifold, with a guarantee on the worst-case distance and the computational complexity. This is a key tool of our analysis; we are not aware of other work that deal with more general (piecewise) polynomial approximation of smooth manifolds.

In Machine Learning, Serra et al. (2018); Hanin & Rolnick (2019); Liu et al. (2023) study the full-dimensional strata of networks built from ReLU activation and linear layers. They provide bounds on the number of full-dimensional strata, as well as a means to compute these strata by a mixed-integer linear program. In contrast, we propose to approximate the tame function by a piecewise polynomial function such that each piece is defined by affine inequalities. Thus, the whole domain is partitioned by affine-hyperplane cuts organized in a hierarchical tree structure, and a polynomial function is fit to each region.

To address *(ii)*, Computational Statistics approximate smooth functions by polynomials on polytopes. There, algorithms are mostly focused on continuous and typically one-dimensional functions (Goldberg et al., 2021; Warwicker & Rebennack, 2021; Jekel & Venter, 2019; Warwicker & Rebennack, 2023), and often consider approximating piecewise linear nonconvex functions (Vielma et al., 2010; Kazda & Li, 2021; Huchette & Vielma, 2023). Piecewise polynomial regression with polynomials of degree $N \geq 2$ is either not addressed Goldberg et al. (2021); Warwicker & Rebennack (2021; 2023) or done through "dimensionality lifting" by appending values of all monomials of degrees 2 to $N$ as extra features to the individual samples Jekel & Venter (2019). This approach of Jekel & Venter (2019) requires us to know the nonsmooth points in advance. We make no such assumption. Function Approximation has a long history of providing theoretical bounds on the distance between the best polynomial approximation of a given $\mathcal{C}^m$ function on a compact set. Jackson-type theorems provide a bound with an explicit dependence to the function to be approximated and the degree of the polynomial, and features the smoothness degree; see e.g., Bagby et al. (2002) for this classical result. Besides, Fefferman (2007) provides a linear operator that extends functions from a set to $\mathbb{R}^n$ while at the same time preserving its regularity degree. Importantly, the operator norm depends only on the space dimension $n$ and the function's regularity, but not on the function itself. These two last result are key to our construction of piecewise polynomial approximation of tame functions.

In Machine Learning, the use of trees befits the task of piecewise polynomial regression, with regression in dimensions higher than 1 utilizing hierarchical partitioning of the input space. In Madrid Padilla et al. (2021), the Dyadic CART algorithm of Donoho (1997) is used to recover a piecewise constant function defined on a lattice of points on the plane, with typical application in image processing and denoising. This builds upon the work on Optimal Regression Trees (ORT) for classification by Bertsimas & Dunn (2017), and is concerned with the optimal axis-aligned partitioning of the space, resulting in a piecewise constant function. We stress that Madrid Padilla et al. (2021) only suggest a brute-force computation of the trees for piecewise constant functions. Notably, Zhang et al. (2023) uses dynamic programming to find the optimal regression tree, for functions that are piecewise constant on axis-aligned strata.

In Statistical Theory, Chatterjee & Goswami (2021) reasons about sample complexity of piecewise polynomial regression via ORT, which (like Madrid Padilla et al. (2021)) assumes that the data is defined over a lattice and that the splits are axis-aligned, but accommodates fitting polynomials of arbitrary degree and lattices of points in arbitrary dimensions $n \geq 2$. The lattice data structure assumption facilitates the possibility of using dynamic programming to solve the mixed-integer program, leading to polynomial-time complexity in the number of samples $n_{\text{samp}}$, but has never been demonstrated in practice. Two obvious shortcomings of the above approaches are that they require both that the data be defined over a regular lattice and that the splits of the tree are axis-aligned. This significantly limits the type of piecewise polynomial functions they can reasonably fit, such as the neural network of Fig. 1, the example function (1) illustrated in Figure 2, or even the simple $\| \cdot \|_\infty$ polyhedral norm. Similar to ORT-LH (Dunn, 2018, Chapter 5) we consider general hyperplane splits, building on the OCT-H formulation. However, ORT-LH considers only piece-wise affine functions, whereas we handle polynomials of arbitrary degree. Additionally, our formulation is linear, while ORT-LH contains $\mathcal{O}(n_{\text{samp}})$ quadratic constraints. This can mean a notable difference in solving speed. A later method, called NNRT Bertsimas et al. (2021), describes a non-optimal approach to learning regression trees by gradient descent. The polynomial functions in leaves are in a non-standard form, i.e., NNRT cannot express all piece-wise polynomial functions. Additionally, the polynomial functions onf NNRT are at most of a degree equal to the tree depth.

**Our contributions** In this work, we combine these results from model theory, approximation theory and optimization theory to present:

- the first theoretical bound on the approximation error of generic *nonsmooth and nonconvex* tame functions by piecewise polynomial functions, see Theorem 1;
- a procedure to compute the best piecewise polynomial function, where each piece is a polyhedron defined by a number of affine inequalities, and the polynomial on each piece has a given degree.

**Notation** $\mathcal{C}^m(A)$ is the set of $m$-times continuously differentiable functions from $A$ to $\mathbb{R}$. We say that $f$ is $\mathcal{C}^m$ (over $A$) when $f \in \mathcal{C}^m(A)$. For a real-valued function $f : A \to \mathbb{R}$, we let $\|f\|_{\infty,A} \triangleq \sup_{x \in A} |f(x)|$. For an integer $n \in \mathbb{N}$, $[n]$ denotes the set of integers from 1 to $n$.

## 2 BACKGROUND ON TAME GEOMETRY

In this section, we outline the main ideas and intuitions of tame geometry, and present the main result of interest: stratification of functions.

### 2.1 DEFINABILITY IN O-MINIMAL STRUCTURES

An o-minimal structure is a collection of certain subsets of $\mathbb{R}^m$, for each $m$, that are stable under a large number of operations. One key property is that one-dimensional sets ($m = 1$) are only given by *finite* unions of intervals and points. The structure is furthermore closed under Boolean operations e.g., taking closures, unions or complements, as well as under projections to lower-dimensional sets, and elementary operations such as addition, multiplication and composition.

**Definition 1** (o-minimal structure). An o-minimal structure on $\mathbb{R}$ is a sequence $\mathcal{S} = (\mathcal{S}_m)_{m \in \mathbb{N}}$ such that for each $m \geq 1$:

 i. $\mathcal{S}_m$ is a boolean algebra of subsets of $\mathbb{R}^m$;
 ii. if $A \in \mathcal{S}_m$, then $\mathbb{R} \times A$ and $A \times \mathbb{R}$ belongs to $\mathcal{S}_{m+1}$;
 iii. $\{(x_1, \ldots, x_m) \in \mathbb{R}^m : x_1 = x_m\} \in \mathcal{S}_m$;
 iv. if $A \in \mathcal{S}_{m+1}$, and $\pi : \mathbb{R}^{m+1} \to \mathbb{R}^m$ is the projection map on the first $m$ coordinates, then $\pi(A) \in \mathcal{S}_m$;
 v. the sets in $\mathcal{S}_1$ are exactly the finite unions of intervals and points.

A set $A \subseteq \mathbb{R}^m$ is said to be *definable* in $\mathcal{S}$ if $A$ belongs to $\mathcal{S}_m$. Similarly, a map $f : A \to B$, with $A \subseteq \mathbb{R}^m$, $B \subseteq \mathbb{R}^n$, is said to be definable in $\mathcal{S}$ if its graph $\Gamma(f) \triangleq \{(x, f(x)) \in \mathbb{R}^{m+n} : x \in A\}$ belongs to $\mathcal{S}_{m+n}$. We will simply use the words definable and *tame* interchangeably to refer to sets and functions that are definable in a given o-minimal structure.

Tame functions are in general nonsmooth and nonconvex, but the tameness properties still make it possible to have some control when studying the behaviour of these functions. The function class is also broad enough to entail most of the cases that would appear in applications across a number of fields. Indeed, in machine learning, most functions that would appear as activation functions in neural networks are tame; see e.g., Bareilles et al. (2025) and references therein.

Due to the balance between being wild enough to include a large number of non-trivial applications while still being tame enough such that it is possible to derive qualitative results on their behaviour, the interest in tame functions has seen a recent surge in numerous fields. In mathematical optimization, this framework allowed showing results that had proved elusive, notably on the convergence to critical points or escaping of saddle points for various (sub)gradient descent algorithms (Davis et al., 2020; Josz, 2023; Bianchi et al., 2023).

### 2.2 TAME FUNCTIONS ARE PIECEWISE $\mathcal{C}^m$

Central results in o-minimality theory are the stratification theorems for sets, and functions. These theorems formalize the fact that tame functions are well-behaved: the domain of a tame function can be partitioned into finitely smooth subsets ($\mathcal{C}^m$ manifolds) on which the function behaves smoothly (it is $\mathcal{C}^m$ on the manifold), with control on how the pieces fit together. We report the statement with the simplest condition on how the subsets fit together van den Dries & Miller (1996); see e.g., Lê Loi (1998); Davis et al. (2025); Bianchi et al. (2023); Bolte et al. (2007) for refinements.

**Proposition 1** (Stratification of functions). *Fix an o-minimal expansion of $\mathbb{R}$, and consider $f : \mathbb{R}^n \to \mathbb{R}$, a definable function, and $m \geq 1$. Then $f$ admits a definable $\mathcal{C}^m$-stratification: there exists a finite partition $\mathcal{W}_m$ of $\mathbb{R}^n$ such that*

• *each stratum $\mathcal{M}$ is a definable connected $\mathcal{C}^m$-manifold,*
• *the restriction of $f$ on each strata $\mathcal{M}$ is $\mathcal{C}^m$-smooth,*
• *any two strata $\mathcal{M}$ and $\mathcal{M}'$ satisfy the frontier condition: $\mathcal{M} \cap \mathrm{cl}\,\mathcal{M}' \neq \emptyset \Rightarrow \mathcal{M} \subset \mathrm{cl}\,\mathcal{M}'$; when the inclusion $\mathcal{M} \subset \mathrm{cl}\,\mathcal{M}'$ holds, $\mathcal{M}$ is said to be* adjacent *to $\mathcal{M}'$.*

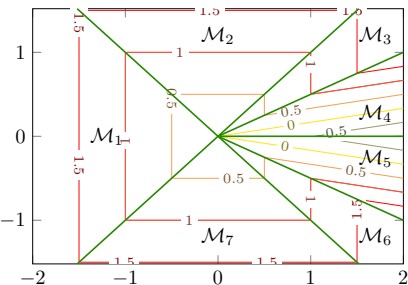

**Figure 2:** Illustration of the "cone" function (1), with $s^{\text{cone}} = r^{\text{cone}} = 0.5$, showing *(i)* the level lines of the function, and *(ii)* the decomposition of the domain into full-dimensional strata on which the function is smooth, as provided by Proposition 1.

**Example 1.** Consider the following tame (piecewise linear) function of $\mathbb{R}^2$:

$$f^{\text{cone}}(x) = \begin{cases} -s^{\text{cone}}x_1 + \frac{1+s^{\text{cone}}}{r^{\text{cone}}}x_2 & \text{if } x_1 > 0 \text{ and } 0 < x_2 < r^{\text{cone}}x_1 \\ -s^{\text{cone}}x_1 - \frac{1+s^{\text{cone}}}{r^{\text{cone}}}x_2 & \text{if } x_1 > 0 \text{ and } -r^{\text{cone}}x_1 < x_2 < 0 \\ \|x\|_\infty & \text{else} \end{cases} \tag{1}$$

When $r^{\text{cone}} = s^{\text{cone}} = 0.5$, as guaranteed by Proposition 1, the domain of the function splits in 7 full-dimensional strata $(\mathcal{M}_i)_{i \in [7]}$, on which the function behaves smoothly. Here the pieces are polytopes, on which the function is linear. Figure 2 shows the landscape of the function, and the collection $\mathcal{W} = (\mathcal{M}_i)_{i \in [7]}$.

## 3  APPROXIMATION OF TAME FUNCTIONS

In this section, we state the main theoretical result: any tame function can be approximated to an arbitrary precision by a piecewise polynomial function.

**Definition 2** ((Piecewise)-polynomial functions)**.** We let $\mathcal{P}_N$ denote the set of polynomials of degree at most $N$. We let $\widetilde{\mathcal{P}}_N^l(A)$ denote the set of functions that are piecewise polynomial on $A$, such that

1. each piece $s$ is a polyhedron defined as the intersection of $l$ halfspaces, represented as one leaf of a complete binary tree of depth $l$ where each node collects an affine split of the space,
2. the restriction of the function to each piece $s$ is a polynomial of degree at most $N$.

**Assumption 1** (On the stratification)**.** The finite stratification $\mathcal{W}_m$ of $A$ is such that, for each stratum $\mathcal{M} \in \mathcal{W}$, there exists two mappings $\Phi_{\mathcal{M}} : \mathbb{R}^n \to \mathbb{R}^{n-\dim(\mathcal{M})}$ and $\Phi_{\partial\mathcal{M}} : \mathbb{R}^n \to \mathbb{R}$ such that

- $\mathcal{M} = \Phi_{\mathcal{M}}^{-1}(\{0\}) \cap \Phi_{\partial\mathcal{M}}^{-1}(\{0\})$,
- zero is a regular point of $\Phi_{\mathcal{M}}$: the Jacobian of $\Phi_{\mathcal{M}}$ has full rank at any point of $\mathcal{M}$,
- $\Phi_{\mathcal{M}}$ is $\mathcal{C}^2$ over $A$; its gradient and Hessian are non-zero and have bounded norm.

We are now ready to state our main result, which combines the stratification property, Proposition 1, with a more classical result from smooth function approximation theory.

**Theorem 1** (Main approximation result)**.** *Consider a function $f : A \to \mathbb{R}$, and a $\mathcal{C}^m$-stratification $\mathcal{W}_m$ of $f$, for some $m \geq 2$, such that:*

- *$f$ is definable in an o-minimal structure,*
- *$A$ is a connected compact subset of $[0,1]^n$, with $n > 1$,*
- *$f$ is continuous over $A$, and the stratification $\mathcal{W}_m$ meets Assumption 1.*

*Then $f$ is approximable by piecewise polynomial functions: for any integers $l \geq 1$, and $N \geq 1$*

$$\inf_{p \in \widetilde{\mathcal{P}}_N^l(A)} \|f - p\|_{\infty,A} \leq (C_1 N^{-m} + C_2 l^{-\frac{2}{n-1}}) \max\left\{\|f\|_{\mathcal{C}^m(\mathcal{M})} : \mathcal{M} \in \mathcal{W}_m \text{ and } \dim(\mathcal{M}) = n\right\}.$$

*where $C_1$ depends only on $n$, $m$, $A$ and $C_2$ depends only on $n$, $m$.*

The definability assumption ensures that the graph of the function does not oscillate arbitrarily, and provides a guarantee on the regularity of the function to be approximated. Specifically, the definability assumption allows us to construct piecewise linear approximations of the set of nonsmooth points of the function, using results from Boissonnat et al. (2023).While it is not clear that the definability assumption is necessary for the results of Boissonnat et al. (2023) to apply, traditional classes of nonsmooth functions, such as weakly convex and $K$-Lipschitz functions, contain functions for which the results do not apply.

Theorem 1 is our main approximation result so, before discussing its proof (which we carry out in detail in Appendix A), some remarks are in order.

- The first term of the bound quantifies the approximation of $f$ on regions of $A$ where $f$ is *smooth*. These regions correspond to the interior of the cells provided by Proposition 1. This term goes to zero as the degree $N$ of the polynomial approximation increases, or as the smoothness $m$ of the underlying function increases. Notably, when $f$ itself is polynomial on the cells and $m$ is high enough, the term $C_1 \propto \|\frac{\partial^m f}{\partial x_j^m}\|_{\infty,\mathrm{diff}_m(f)}$ vanishes and the polynomial approximant matches $f$ exactly.
- The second term of the bound quantifies the approximation of $f$ on *nonsmooth* regions. This term goes to zero as the size of the pieces of the polynomial that contain nondifferentiability points of $f$ goes to zero. This is ensured by increasing the number of pieces $k$.

**Proof outline of Theorem 1.** The result is obtained by constructing a specific piecewise polynomial function that satisfies the bound. To do so, we first consider the set of points where the function is not $\mathcal{C}^m$. The stratification result Proposition 1 guarantees that this set decomposes into a finite union of $\mathcal{C}^m$ manifolds of dimension at most $n-1$. The result of Boissonnat et al. (2023) then provides a piecewise linear approximation with a controlled approximation error depending on the number of prescribed linear pieces, for each manifold. We thus consider the subset of piecewise polynomials for which the linear cuts yield exactly the linear spaces defined above. The problem thus reduces to finding the best degree $N$ polynomial approximation of $f$ on each of the polyhedron. Two situations emerge.

(i) Either $f$ is $\mathcal{C}^m$ on the given polyhedron, in which case classical results of smooth approximation theory (e.g., Jackson's theorem) provide a suitable degree $m$ polynomial approximation and a corresponding bound on the error, or

(ii) Either $f$ is nonsmooth on the polyhedron. In that case, we consider a simple linear approximation of the function on the polyhedron and bound the error using Lipschitz contiunuity of the function.

*Remark* 1 (Extensions to non-cubic domains). The domain $A$ on which $f$ is approximated is a full-dimensional cube. However, one might use approximation of smooth multivariate functions on more general domains, e.g., polyhedral or more general (Totik, 2020; Dai & Prymak, 2023).

## 4 NUMERICS

In this section, we briefly address the problem of piecewise polynomial regression as a mixed-integer program (MIP), followed by a discussion of experimental evaluation.

### 4.1 PIECEWISE POLYNOMIAL APPROXIMATION USING MIP

Our MIP formulation is based on the optimal classification trees framework (Bertsimas & Dunn, 2017). In short, we utilize a modification of the affine-hyperplane tree structure proposed by Bertsimas & Dunn (2017) under the name OCT-H to separate the input space into polyhedral regions. The most notable modification to the formulation is in the splitting functions, which take the form of $a_m^\top x_i \geq b_m$ for sample $i$ and branching node $m$. Unlike Bertsimas & Dunn (2017), we enforce that $\|a_m\|_1 = 1$. In the OCT-H, the norm of $a_m$ is constrained to be *at most* 1. Our formulation greatly reduces the number of equivalent solutions by making $(a_m/z)^\top x_i \geq b_m/z$ for any $z > 1$ infeasible, unlike in the OCT-H formulation (assuming $\|a_m\|_1 = 1$). Empirical evidence suggests that the equality constraint leads to faster improvements of the primal bound, i.e., finding high-quality solutions faster.

Based on the depth $D$ of the tree, we create $2^D$ polyhedral regions in leaves. For samples belonging to each of the regions (leaves), we formulate a Mean absolute error (MAE) with respect to some polynomial function of degree $N$. We also require at least $N_{\min}$ samples in each non-empty leaf to ensure a reasonable fit of the polynomial function.

We enable the use of polynomial functions in the regions straightforwardly by introducing an extra attribute (lifted feature) for each monomial and then fitting a linear function on all such attributes. We linearize the MAE objective by introducing proxy variables $\delta_i$ for each sample $i$ and bounding them from below by the relative error and its negation. This is valid since we minimize a weighted sum of the values. We provide the full details of the formulation in Eq. (12) (Appendix B).

*Remark* 2 (Axis-aligned regression). In Appendix C, we propose a version of (12) tailored to functions whose full-dimensional strata have boundaries aligned with cartesian axes. Such functions appear in signal processing applications, and are the topic of recent works Chatterjee & Goswami (2021); Bertsimas & Dunn (2017); Madrid Padilla et al. (2021).

## 4.2 EXPERIMENTS

In this section, we demonstrate the applicability of the piecewise polynomial regression method developed in the previous section. Appendix D contain complementary details and experiments.

**Setup**  We implement the affine-hyperplane formulation, detailed in Eq. (12) in Appendix B, in Python and use Gurobi as the MIP solver. We present three applications: regression of the cone function (1), regression of a Neural Network, and denoising of a piecewise constant 2d signal, following the experiments of (Madrid Padilla et al., 2021). We use trees of varying depth $D \in \{2, 3, 4\}$ and polynomial degree $N \in \{0, 1, 2\}$. We set $N_{\min} = 1$ and $\mu = 10^{-4}$ across all experiments, and run experiments on a personal laptop with CPU AMD Ryzen 7 PRO 6850U with 8 Cores (16 Logical Processors) and 32GB of RAM, unless specified otherwise. All methods share the minimum number of points in a partition $N_{\min} = 1$ and a small constant $\mu = 10^{-4}$, used to formulate strict inequalities in (12).

### 4.2.1 REGRESSION OF TAME FUNCTIONS

We consider the regression problem for two tame functions and report in Fig. 3 and Fig. 4 *i.* the level lines of the original function and its piecewise polynomial approximation, and *ii.* the relative approximation error, defined as the approximation error normalized by the maximal absolute value of the approximate function on the domain. Table 1 shows the error computed on a $10^3 \times 10^3$ grid of regularly spaced points.

**Piecewise-affine function**  Firstly, we return to the "cone" function. This function is piecewise-linear and continuous; see Fig. 3a (left pane) for an illustration. This is a nonconvex function for which estimating the correct full-dimensional strata with a sampling scheme gets harder as $r^{\text{cone}}$ goes to zero or as the dimension of the space $n$ increases.

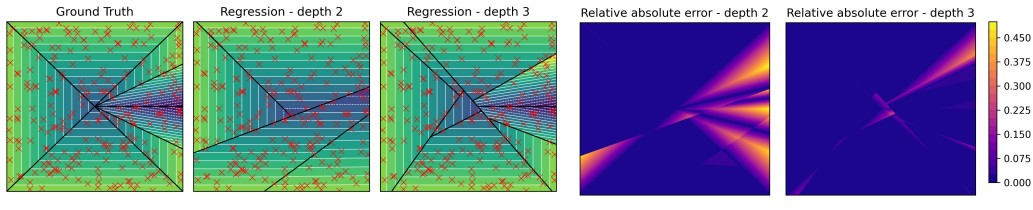

(a) Ground truth and regression results.                    (b) Relative approximation error.

**Figure 3:** Approximation of the cone function Eq. (1) by trees of depth 2 and 3. Black lines show the boundaries of the full-dimensional strata; red crosses show the samples used for building the approximation. The relative error is computed as the absolute error divided by the maximal ground truth value in the shown function domain.

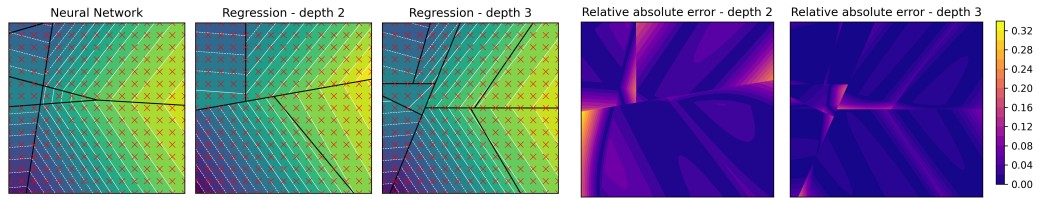

**(a)** Ground truth and regression results.      **(b)** Relative approximation error.

**Figure 4:** Approximation of the neural network (Fig. 5) by trees of depth 2 and 3. Black lines show the boundaries of the full-dimensional strata; red crosses show the samples used for building the approximation. The relative error is computed as the absolute error divided by the maximal ground truth value in the shown function domain.

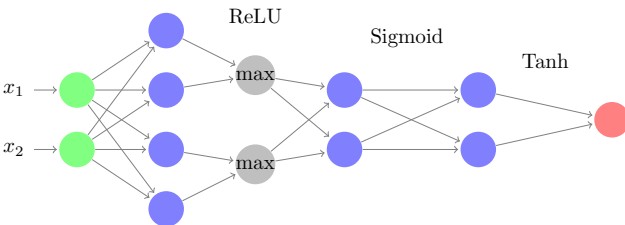

**Figure 5:** The architecture of the example Neural Network. Input neurons are green, blue nodes are neurons in hidden layers, and the output neuron is red. Grey nodes represent the max-pooling layer. Above the connections are the names of activation functions used on the outputs of the layer to the left of each respective name. The results using this architecture are in Figure 4 and Table 1.

Figure 3a (middle and right pane) shows the obtained piecewise approximation of $f^{\mathrm{cone}}$ (with $r^{\mathrm{cone}} = s^{\mathrm{cone}} = 0.5$), for depth 2 and 3 approximation trees with time budget of 5 and 10 minutes. There are $n_{\mathrm{samp}} = 250$ points sampled uniformly; the polynomial degree is $N = 1$. We can inspect the approximation error closer in Fig. 3b. The error is highest in areas with inaccurate partitioning. The depth $D = 2$ tree fails to recover an approximation of the strata, as it can only approximate 4 full-dimensional strata while $f^{\mathrm{cone}}$ has 7. The depth $D = 3$ recovers the stratification well qualitatively, and reduces error by a factor 2; see Table 1.

**Non-piecewise-linear Neural Network** We consider a small neural network, comprised of 27 parameters, sigmoid, `tanh`, ReLU activations, and 1d max-pooling; see the architecture in Figure 5. The neural network is trained to approximate the 2d function $x \mapsto 2\sin x_1 + 2\cos x_2 + x_2/2$, from 15 random samples on the cube $[-2, 2]^2$. The loss is the mean squared error over the 15 samples, optimized in a single batch for 5000 epochs using the `AdamW` algorithm.

We approximate the trained NN by taking $n_{\mathrm{samp}} = 225 = 15^2$ points in a $15 \times 15$ regular grid over the approximation space and optimize the trees using that as input. We set the polynomial degree $N = 2$.

Figure 4a (middle and right pane) shows the obtained piecewise approximation of the Neural Network for depth 2 and 3 approximation trees with a time budget of 30 and 60 minutes. We can inspect the areas with high error in Fig. 4b. Notice the non-linearity of the errors, caused by utilization of degree 2 polynomial functions. Increasing the depth from 2 to 3 reduces again the error measures by a factor of 2; see Table 1.

### 4.2.2 Notes on the optimization process

To study the MIP optimization process until completion, we also consider $f = \|\cdot\|_\infty$ in addition to the cone function Eq. (1) and the neural network. The norm is a simple tame function whose stratification cannot be approximated precisely by trees with axis-aligned splits; see e.g., the recent works Madrid Padilla et al. (2021); Chatterjee & Goswami (2021).

**Table 1:** Normalized error between tame functions and affine-hyperplane tree approximations (12).

| Function | depth | max. err. | mean err. | median err. |
|---|---|---|---|---|
| $f^{\text{cone}}$ (1) | 2 | $5.0 \times 10^{-1}$ | $4.7 \times 10^{-2}$ | $1.0 \times 10^{-4}$ |
| | 3 | $3.7 \times 10^{-1}$ | $1.0 \times 10^{-2}$ | $6.4 \times 10^{-5}$ |
| NN | 2 | $3.3 \times 10^{-1}$ | $3.6 \times 10^{-2}$ | $2.5 \times 10^{-2}$ |
| | 3 | $1.9 \times 10^{-1}$ | $1.3 \times 10^{-2}$ | $9.3 \times 10^{-3}$ |

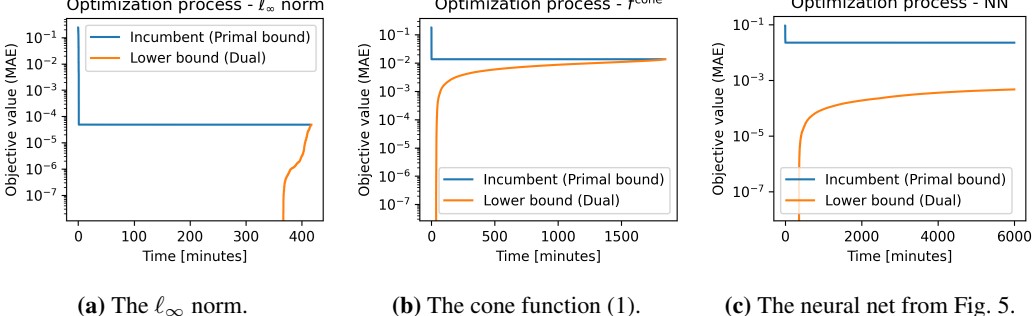

**(a)** The $\ell_\infty$ norm.  **(b)** The cone function (1).  **(c)** The neural net from Fig. 5.

**Figure 6:** The optimization process for various tame functions. Objective value (mean absolute error) of the incumbent solution during the optimization. The full optimization takes hours, although a solution with almost optimal error is found within tens of seconds. The neural network optimization did not prove optimality within 100 hours.

We approximate the cone function and the norm over $n_{\text{samp}} = 100$ randomly sampled points with depth $D = 2$ and polynomial degree $N = 1$. For the neural network, the setup is the same as in Section 4.2.1, i.e., learning trees on 255 samples in a grid with depth $D = 2$ and polynomial degree $N = 2$. These experiments were run for up to 100 hours on an internal cluster with 8 cores and 16GB RAM.

Figure 6 shows the evolution of objective value bounds: the blue curve shows the objective value of the best candidate found by the solver so far, the orange curve shows the best lower bound on the optimal objective value found so far. A candidate piecewise polynomial function is proved to be optimal when the two curves coincide. For the norm (Fig. 6a) and the cone function (Fig. 6b), we prove optimality within the time limit. It takes a couple of hours, but the incumbent solution has the objective value within $10^{-6}$ of the global optimum already after 29 and 65 seconds for the cone function and the norm, respectively. Most of the effort in solving (12) is thus spent on proving optimality of the candidate function. This is an even more extreme contrast between the optimization of the two bounds as compared to the results for OCT Bertsimas & Dunn (2017). This suggests that although the neural network (Fig. 6c) and the rest of our results were not proven optimal, the solutions could be close to optimal after a short computing time.

## 5  CONCLUSIONS

Our numerical results showcase a promising proof-of-concept implementation, whose scalability (in terms of proving global optimality) is limited to low-dimensional functions, while a good feasible solution can be found surprisingly quickly; see Fig. 6. Improvement in the search for lower (dual) bounds is an important direction of future work.

Most importantly, we present and prove the first theoretical bound on the approximation error of generic tame functions (including functions represented with neural networks) by piece-wise polynomial functions.

REPRODUCIBILITY STATEMENT

Code to reproduce the experiments is provided in the Supplementary Material. This includes a readme file with detailed instructions to reproduce experiments. Details on the computing environment are provided as the last paragraph of Section 4.2.

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

## A    APPROXIMATION RESULTS WITH PROOF

In this section, we prove the main Theorem 1: any tame function can be approximated to an arbitrary precision by a piecewise polynomial function. We first provide preliminary results in Appendix A.1, and proceed with the proof in Appendix A.2.

### A.1    PRELIMINARY RESULTS

We follow the standard notion of the differential $\mathrm{D}^\gamma f$ of a function $f : \mathbb{R}^n \to \mathbb{R}$, for a multi-index $\gamma \in \mathbb{R}^n$. Following Fefferman (2007), we denote by $\mathcal{C}^m(\mathbb{R}^n)$ the space of real-valued functions on $\mathbb{R}^n$ with continuous and bounded derivatives through order $m$, and let, for $f \in \mathcal{C}^m(\mathbb{R}^n)$,

$$\|f\|_{\mathcal{C}^m(\mathbb{R}^n)} \triangleq \max_{|\gamma| \le m} \sup_{x \in \mathbb{R}^n} |\mathrm{D}^\gamma f(x)|.$$

Furthermore, for a subset $E$ of $\mathbb{R}^n$, we denote by $\mathcal{C}^m(E)$ the Banach space of all real-valued functions $\varphi$ on $E$ such that $\varphi = f$ on $E$ for some $f \in \mathcal{C}^m(\mathbb{R}^n)$. The norm on $\mathcal{C}^m(E)$ is

$$\|\varphi\|_{\mathcal{C}^m(E)} \triangleq \inf_{f \in \mathcal{C}^m(\mathbb{R}^n) \text{ and } f=\varphi \text{ on } E} \|f\|_{\mathcal{C}^m(\mathbb{R}^n)}.$$

Finally, the Frechet distance $d_F(\cdot, \cdot)$ between two homeomorphic manifolds is

$$d_F(\mathcal{M}_a, \mathcal{M}_b) \triangleq \inf_{h \in \mathcal{H}} \sup_{x \in \mathcal{M}_a} d(x, h(x)),$$

where $\mathcal{H}$ is the set of all homemorphisms from $\mathcal{M}_a$ to $\mathcal{M}_b$.

### A.1.1 SMOOTH EXTENSION OF FUNCTIONS

We will extend regular functions from subsets of $A$ to $\mathbb{R}^n$. The following result, Theorem 1 in Fefferman (2007), allows doing so while controlling the norm of the extended function.

**Proposition 2.** *Let $E \subset \mathbb{R}^n$ and $m \geq 1$. Then, there exists a linear map $T : \mathcal{C}^m(E) \to \mathcal{C}^m(\mathbb{R}^n)$ such that $T(f) = f$ on $E$, for each $f \in \mathcal{C}^m(E)$, and the norm of $T$ is bounded by a constant depending only on $n$ and $m$.*

### A.1.2 SMOOTH MULTIVARIATE POLYNOMIAL APPROXIMATION

We will employ the following polynomial approximation result, Theorem 2 in Bagby et al. (2002).

**Proposition 3.** *Let $K$ be a connected subset of $\mathbb{R}^n$ such that any two points $a$ and $b$ can be joined by a rectifiable arc of length no greater than $\sigma \|a - b\|$, where $\sigma$ is a positive constant. Let $f$ be a $\mathcal{C}^m$ function on an open neighborhood of $K$, where $0 \leq m < \infty$. Then, for each nonnegative integer $N$, there exists a polynomial $p_N$ of degree at most $N$ such that, for each multi-index $\alpha$ with $|\alpha| \leq \min(n, N)$ we have*

$$\sup_K |\mathrm{D}^\alpha(f - p_N)| \leq \frac{C_{n,m,K}}{N^{m-|\alpha|}} \sum_{|\gamma| \leq m} \sup_K |\mathrm{D}^\gamma f|,$$

*where $C_{n,m,K}$ is a positive constant depending only on $N$, $m$, and $K$.*

### A.1.3 PIECEWISE-LINEAR APPROXIMATION OF THE STRATIFICATION

**Proposition 4.** *Consider a stratification $\mathcal{W}$ of $\mathbb{R}^n$, $n > 1$, definable in an o-minimal expansion of $\mathbb{R}$ that meets Assumption 1. There exists a binary tree partition of $A$ with depth $l$, as defined in Definition 2, such that the distance between each codimension 1 strata, $\mathcal{M}$, and the corresponding partition boundary, $\widehat{\mathcal{M}}$, is bounded by*

$$d_F(\mathcal{M}, \widehat{\mathcal{M}}) \leq C^3 l^{-\frac{2}{n-1}},$$

*where the constant $C^3$ depends only on the dimension $n$ and the geometry of the strata $\mathcal{M}$.*

*Proof.* We apply the algorithm of Boissonnat et al. (2023) to the closure of the strata $\mathcal{M}$. From (Boissonnat et al., 2023, Thm. 3.4) we have that $\mathrm{dist}(\mathcal{M}, \widehat{\mathcal{M}}) \leq C^4 D^2$, where $D$ is the maximal diameter of a linear piece. Here the constant depends on the magnitude of the gradient and Hessian of the mapping $\Phi_{\mathcal{M}}$ from Assumption 1. From (Boissonnat et al., 2023, Prop. 3.6) we have $l \leq C^5 D^{-(n-1)}$, where now the constant $C^5$ depends on the space dimension $n$, and the number of times any straight line intersects $\mathcal{M}$. Note that this number is finite by the definability assumption, as is well-known in the case of algebraic manifolds. Putting this together gives the result. ∎

### A.2 PROOF OF THEOREM 1

The result is obtained by considering a piecewise linear approximation of the nonsmooth regions of the function constructed by intersecting $l$ halfplanes. The proof then splits into two parts, which correspond to the two terms in the bound in Theorem 1.

*Proof of Theorem 1.* The proof consists in constructing a piecewise polynomial function $\hat{p}$ in $\widetilde{\mathcal{P}}_N^l$ that has the claimed distance to $f$.

We first construct the depth-$l$ binary tree that defines the pieces of the piecewise polynomial. Since $f$ is definable in an o-minimal structure, Proposition 1 yields a $\mathcal{C}^m$-decomposition $\mathcal{W}$ of $A$ such that *i.* each stratum is a connected $\mathcal{C}^m$-manifold, and *ii.* the restriction of $f$ to each stratum $\mathcal{M}$ is $\mathcal{C}^m(\mathcal{M})$.

If the decomposition meets Assumption 1, Proposition 4 provides a way to recursively split the space along affine hyperplanes such that, for any stratum, $\mathcal{M}$ that is not full-dimensional and any point $x$ in $\mathcal{M}$, there exists a point of the boundary between the pieces that is at most $C^3 l^{-\frac{2}{n-1}}$ away from $x$. This specifies exactly the geometry of the pieces of the piecewise polynomial function we construct. For conciseness, we also let

$$\bar{C}_f \triangleq \max \left\{ \|f\|_{\mathcal{C}^m(\mathcal{M})} : \mathcal{M} \in \mathcal{W} \text{ and } \dim(\mathcal{M}) = n \right\}.$$

Consider a full-dimensional piece $s \subset A$ of the recursive splitting of $A$ defined above. We now turn to define a polynomial that approximates $f$ on $s$. That polynomial will define $\hat{p}|_s$, the restriction of the piecewise polynomial function to $s$. First, note that by construction of $s$, all points of $s$ that are more than $C^3 l^{-\frac{2}{n-1}}$ away from the set of nondifferentiable points of $f$ belong to the same full-dimensional stratum. Let $\mathcal{M}_0^s$ denote such a stratum if it exists, or otherwise, an arbitrary full-dimensional stratum that intersects with $s$.

By construction of the stratification, $f$ is $\mathcal{C}^m$ on $\mathcal{M}_0^s$, thus in particular on the subset $\mathcal{M}_0^s \cap s$. Proposition 2 therefore provides $T(f) \in \mathcal{C}^m(\mathbb{R}^n)$, a smooth extension of $f$ from $\mathcal{M}_0^s \cap s$ to $\mathbb{R}^n$, and a constant $C_{n,m}^1$ that depends only on $n$ and $m$, such that

$$\|T(f)\|_{\mathcal{C}^m(\mathbb{R}^n)} \leq C_{n,m}^1 \|f\|_{\mathcal{C}^m(\mathcal{M}_0^s \cap s)}. \tag{2}$$

As the domain of a definable function, $A$ is also definable, and thus naturally meets the assumption that any two points $x$ and $y$ in $A$ can be joined by a rectifiable arc in $A$ with length no greater than $\sigma \|x - y\|$, where $\sigma$ is a positive constant. Besides, $A$ is a connected compact set by assumption, and $T(f)$ is $\mathcal{C}^m$ on $\mathbb{R}^n$. Proposition 3 therefore provides the existence of a polynomial $p_N^s(x)$ of degree up to $N$ that approximates $T(f)$ on $A$:

$$\|T(f) - p_N^s(x)\|_{\infty, A} \leq C_{n,m,A}^2 \frac{1}{N^m} \sum_{|\gamma| \leq m} |\operatorname{D}^\gamma T(f)|_{\infty, A}. \tag{3}$$

We let $\hat{p}|_s \triangleq p_N^s$, and proceed to bound the approximation error. This error splits in two terms: for any $x \in s$, the triangular inequality implies

$$|f(x) - p_N^s(x)| \leq |f(x) - T(f)(x)| + |T(f)(x) - p_N^s(x)|. \tag{4}$$

The first term of the right hand side of (4) admits the following bound

$$|f(x) - T(f)(x)| \leq n \left( \bar{C}_f + \|T(f)\|_{\mathcal{C}^m(\mathbb{R}^n)} \right) \|x - \pi_{\mathcal{M}_0^s \cap s}(x)\|. \tag{5}$$

where $\pi_{\mathcal{M}_0^s \cap s}(x)$ denotes the orthogonal projection of $x$ onto $\mathcal{M}_0^s \cap s$. Indeed, we first have

$$
\begin{aligned}
|f(x) - T(f)(x)| \leq &|f(x) - f(\pi_{\mathcal{M}_0^s \cap s}(x))| \\
&+ |f(\pi_{\mathcal{M}_0^s \cap s}(x)) - T(f)(\pi_{\mathcal{M}_0^s \cap s}(x))| + |T(f)(\pi_{\mathcal{M}_0^s \cap s}(x)) - T(f)(x)|.
\end{aligned}
$$

Since $f$ admits a finite definable stratification, there exists a family $t_0 = 0 < ... < t_{k+1} = 1$ such that each open segment $I_i = (x + t_i(\pi_{\mathcal{M}_0^s \cap s}(x) - x), x + t_{i+1}(\pi_{\mathcal{M}_0^s \cap s}(x) - x))$, for $0 \leq i \leq k$, belongs to one full-dimensional stratum $\mathcal{M}_i$ of $\mathcal{W}$. Note that, since the stratification is definable, there cannot exist a segment $[a, b] \subset A$ that intersects infinitely many strata. Therefore, the size of the family $(t_i)_{0 \leq i \leq k}$ is finite. Let $\varphi(t) = f(x + t(\pi_{\mathcal{M}_0^s \cap s}(x) - x))$. Since $f$ is $\mathcal{C}^m$ on each open segment $I_i$, continuous over $A$, and $I_i \subset \mathcal{M}_i$

$$\varphi(t_i) - \varphi(t_{i+1}) \leq (t_{i+1} - t_i) \sup_{u \in \mathcal{M}_i} \|\nabla f(u)\| \|x - \pi_{\mathcal{M}_0^s \cap s}(x)\|.$$

Combining the inequality $\| \cdot \|_2 \leq n \| \cdot \|_\infty$ and the definition of the norm on the Banach space $\mathcal{C}^m(\mathcal{M}_i)$, we have

$$\sup_{u \in \mathcal{M}_i} \|\nabla f(u)\| \leq n \|f\|_{\mathcal{C}^m(\mathcal{M}_i)} \leq n \max \left\{ \|f\|_{\mathcal{C}^m(\mathcal{M})} : \mathcal{M} \in \mathcal{W} \text{ and } \dim(\mathcal{M}) = n \right\}$$

Thus, since $\bar{C}_f = \max \left\{ \|f\|_{\mathcal{C}^m(\mathcal{M})} : \mathcal{M} \in \mathcal{W} \text{ and } \dim(\mathcal{M}) = n \right\}$, there holds

$$f(x) - f(\pi_{\mathcal{M}_0^s \cap s}(x)) = \sum_{i=0}^{k} \varphi(t_i) - \varphi(t_{i+1}) \leq n \bar{C}_f \|x - \pi_{\mathcal{M}_0^s \cap s}(x)\|. \tag{6}$$

Besides, since $T(f)$ is an extension of $f$ from $s \cap \mathcal{M}_0^s$ to $\mathbb{R}^n$, and $\pi_{\mathcal{M}_0^s \cap s}(x)$ belongs to the closure of $s \cap \mathcal{M}_0^s$, there holds $f(\pi_{\mathcal{M}_0^s \cap s}(x)) = T(f)(\pi_{\mathcal{M}_0^s \cap s}(x))$. Finally, since $T(f) \in \mathcal{C}^m(\mathbb{R}^n)$,

$$|T(f)(\pi_{\mathcal{M}_0^s \cap s}(x)) - T(f)(x)| \leq \sup_{y \in \mathbb{R}^n} \|\nabla T(f)(y)\| \|x - \pi_{\mathcal{M}_0^s \cap s}(x)\|. \tag{7}$$

Combining the inequality $\| \cdot \|_2 \leq n \| \cdot \|_\infty$ with the norm on $\mathcal{C}^m(\mathbb{R}^n)$ yields

$$\sup_{y \in \mathbb{R}^n} \|\nabla T(f)(y)\| \leq n \|T(f)\|_{\mathcal{C}^m(\mathbb{R}^n)}. \tag{8}$$

Finally, combining Eqs. (6)–(8) yields the claimed bound Eq. (5).

The second term on the right hand side of (4) admits the following bound

$$\|T(f) - p_N^s(x)\|_{\infty, A} \leq C_{n,m,A} \|T(f)\|_{\mathcal{C}^m(\mathbb{R}^n)} N^{-m}. \tag{9}$$

Indeed, note that, for any multi-index $\gamma$ such that $|\gamma| \leq m$,

$$|\operatorname{D}^\gamma T(f)|_{\infty, A} \leq |\operatorname{D}^\gamma T(f)|_{\infty, \mathbb{R}^n} \leq \max_{|\gamma| \leq m} |\operatorname{D}^\gamma T(f)|_{\infty, \mathbb{R}^n},$$

so that

$$\sum_{|\gamma| \leq m} |\operatorname{D}^\gamma T(f)|_{\infty, A} \leq \binom{m+n}{n} \max_{|\gamma| \leq m} |\operatorname{D}^\gamma T(f)|_{\infty, \mathbb{R}^n} = \binom{m+n}{n} \|T(f)\|_{\mathcal{C}^m(\mathbb{R}^n)}.$$

Letting $C_{n,m,A} = C_{n,m,A}^2 \binom{m+n}{n}$, and combining the above equation and Eq. (3) provides Eq. (9).

Combining Eqs. (5) and (9) thus yields a bound on the approximation error: for all $x \in s$

$$|f(x) - p_N^s(x)| \leq n \left( \bar{C}_f + \|T(f)\|_{\mathcal{C}^m(\mathbb{R}^n)} \right) \|x - \pi_{\mathcal{M}_0^s \cap s}(x)\| + C_{n,m,A} \|T(f)\|_{\mathcal{C}^m(\mathbb{R}^n)} N^{-m}$$

Using that $\|x - \pi_{\mathcal{M}_0^s \cap s}(x)\| \leq C^3 l^{-\frac{2}{n-1}}$ for $x \in s$, and Eq. (2) to control the norm of the smooth extension, there holds for all $x \in s$,

$$|f(x) - p_N^s(x)| \leq n(\bar{C}_f + C_{n,m}^1 \|f\|_{\mathcal{C}^m(\mathcal{M}_0^s \cap s)}) C^3 l^{-\frac{2}{n-1}} + C_{n,m,A} C_{n,m}^1 \|f\|_{\mathcal{C}^m(\mathcal{M}_0^s \cap s)} N^{-m} \tag{10}$$

Since $s \cap \mathcal{M}_0^s \subset \mathcal{M}_0^s$, any function $\varphi \in \mathcal{C}^m(\mathbb{R}^n)$ that matches $f$ on $\mathcal{M}_0^s$ also matches $f$ on $s \cap \mathcal{M}_0^s$, so that

$$\|f\|_{\mathcal{C}^m(s \cap \mathcal{M}_0^s)} = \inf \left\{ \|\varphi\|_{\mathcal{C}^m(\mathbb{R}^n)} : \varphi \in \mathcal{C}^m(\mathbb{R}^n) \text{ and } \varphi = f \text{ on } s \cap \mathcal{M}_0^s \right\}$$

$$\leq \inf \left\{ \|\varphi\|_{\mathcal{C}^m(\mathbb{R}^n)} : \varphi \in \mathcal{C}^m(\mathbb{R}^n) \text{ and } \varphi = f \text{ on } \mathcal{M}_0^s \right\} = \|f\|_{\mathcal{C}^m(\mathcal{M}_0^s)}.$$

Thus,

$$\|f\|_{\mathcal{C}^m(s \cap \mathcal{M}_0^s)} \leq \|f\|_{\mathcal{C}^m(\mathcal{M}_0^s)} \leq \max \left\{ \|f\|_{\mathcal{C}^m(\mathcal{M})} : \mathcal{M} \in \mathcal{W} \text{ and } \dim(\mathcal{M}) = n \right\} = \bar{C}_f,$$

so that the bound of (10) is now independent of $s$:

$$|f(x) - p_N^s(x)| \leq n \bar{C}_f (1 + C_{n,m}^1) C^3 l^{-\frac{2}{n-1}} + C_{n,m,A} C_{n,m}^1 \bar{C}_f N^{-m}$$

Taking the supremum over all $x$ in $s$, and then over all pieces $s$ of $A$ yields the bound for the piecewise polynomial function $\hat{p}$

$$\|f - p\|_{A,\infty} \leq (C_1 N^{-m} + C_2 l^{-\frac{2}{n-1}}) \bar{C}_f.$$

where $C_1 = C_{n,m,A} C_{n,m}^1$ depends only on $n$, $m$, $A$, and $C_2 = n(1 + C_{n,m}^1) C^3$ depends only on $n$, and $m$. ∎

## B   MIP FORMULATION OF THE REGRESSION

In this section, we formulate the problem of piecewise polynomial regression as a mixed-integer program (MIP), inspired by the optimal classification trees framework (Bertsimas & Dunn, 2017).

The mixed-integer optimization problem expects as input $n_{\text{samp}}$ points $(x_i)_{i \in [n_{\text{samp}}]}$ that belong to $A$, and the corresponding function values $y_i = f(x_i)$ for $i \in [n_{\text{samp}}]$. Without loss of generality, we assume that the sample points belong to $[0,1]^n$. The output is a piecewise polynomial function; the boundaries of the smooth pieces are defined by affine hyperplanes. Tables 3 and 4 summarize the hyperparameters and variables of the mixed-integer formulation; we now explain the formulation details.

**Binary tree**   We consider a fixed binary tree of depth $D$. The tree has $T = 2^{D+1} - 1$ nodes, indexed by $t = 1, \dots, T$ such that all branching nodes are indexed by $t = 1, \dots, 2^D - 1$ and leaf nodes are indexed by $t = 2^D, \dots, 2^{D+1} - 1$. The sets of branching nodes and leaf nodes are denoted $\mathcal{T}_B$ and $\mathcal{T}_L$ respectively. Besides, the set of ancestors of node $t$ is denoted $A(t)$. This set is partitioned into $A_L(t)$ and $A_R(t)$, the subsets of ancestors at which branching was performed on the left and right respectively. Figure 7 shows a tree of depth $D = 2$ where e.g., the ancestors of node 6 are $A(6) = \{1, 3\}$, and left and right branching ancestors are $A_L = \{3\}$ and $A_R = \{1\}$. Each leaf corresponds to an element of the partition, defined by the inequalities of its ancestors e.g., leaf 6 corresponds to $\{x \in \mathbb{R}^n : a_1^\top x \geq b_1, a_3^\top x < b_3\}$.

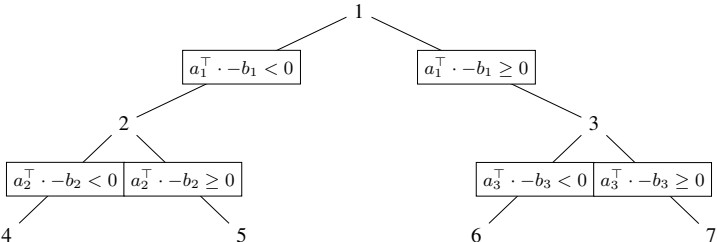

**Figure 7:** Binary tree and corresponding partition.

**Affine-hyperplane partition of the space**   At each branching node $m \in \mathcal{T}_B$, a hyperplane splits the space in two subspaces

$$a_m^\top x_i - b_m < 0 \qquad a_m^\top x_i - b_m \geq 0, \tag{11}$$

that will be associated to the left and right children of node $m$. The parameters $a_m \in \mathbb{R}^n$ and $b_m \in \mathbb{R}$ are variables of the mixed integer program.

Making this formulation practical requires two precisions. First, in order to avoid scaling issues, we constrain $a_m$ to belong in $[-1, 1]^n$, such that $\|a_m\|_1 = 1$.[1] This formulates as

$$
\begin{aligned}
\sum_{j=1}^d \left( a_{mj}^+ + a_{mj}^- \right) &= 1 & \\
a_{mj} &= a_{mj}^+ - a_{mj}^- & \forall j \in [n] \\
a_{mj}^+ &\leq o_{mj} & \forall j \in [n] \\
a_{mj}^- &\leq (1 - o_{mj}) & \forall j \in [n]
\end{aligned}
$$

Furthermore, since $x_i \in [0, 1]^n$, it holds that $a_m^\top x_i \in [-1, 1]$, so that $b_m$ is constrained to $[0, 1]$ without loss of generality. Second, we implement the strict inequality (11) by introducing variable $z_{ti}$ and a small constant $\mu > 0$. Noting that $a_m^\top x_i - b_m$ takes values in the interval $[-2, 2]$, the affine split inequalities (11) now formulate as

$$
\begin{aligned}
a_m^\top x_i &\geq b_m - 2(1 - z_{ti}) & \forall m \in A_R(t) \\
a_m^\top x_i + \mu &\leq b_m + (2 + \mu)(1 - z_{ti}) & \forall m \in A_L(t)
\end{aligned}
$$

both for all $i$ in $[n_{\text{samp}}]$ and all $t$ in $\mathcal{T}_L$.

**Regression on each partition element**   Each leaf $t \in \mathcal{T}_L$ corresponds to one element of the partition of $A$ defined by the tree, and is associated with a degree $r$ polynomial $\text{poly}(\cdot; c_t)$ whose coefficients are parameterized by variable $c_t$. The regression error of $(x_i, y_i$ by the polynomial of leaf $t$ is then $\phi_{ti} = y_i - \text{poly}(x_i; c_t)$. Each point $x_i$ is assigned to a unique leaf of the tree: for all $i \in [n_{\text{samp}}]$, $\sum_{t \in \mathcal{T}_L} z_{ti} = 1$. Furthermore, each leaf $t \in \mathcal{T}_L$ is either assigned zero or at least $N_{min}$ points: for all $i \in [n_{\text{samp}}]$, $\sum_{i=1}^{n_{\text{samp}}} z_{ti} \geq N_{\min} l_t$ and $z_{ti} \leq l_t$.

---

[1] We depart here from the OCT-H formulation Bertsimas & Dunn (2017), which constrains the norm of $a_m$ to be at most 1. This is due to empirical evidence suggesting that equality constraint leads to faster improvements of the primal bound.

**Minimizing the regression error**  The objective is minimizing the average prediction error $\sum_{i=1}^{n_{\text{samp}}} |y_i - \text{poly}(x_i; c_{t(i)})|$, where $c_{t(i)}$ is the polynomial coefficients corresponding to the leaf to which $x_i$ belongs. We formulate this as minimizing $n_{\text{samp}}^{-1} \sum_{i=1}^{n_{\text{samp}}} \delta_i$, where $\delta_i$ models the absolute value of the prediction error for point $x_i$:

$$\delta_i \geq \phi_{ti} - (1 - z_{ti})M \quad \text{for all } t \in \mathcal{T}_L$$
$$\delta_i \geq -\phi_{ti} - (1 - z_{ti})M \quad \text{for all } t \in \mathcal{T}_L$$

where $M$ is a big constant that makes the constraint inactive when $z_{ti} = 0$. Note that the above big-$M$ formulation can be replaced by indicator constraints, if the solver supports them, to encode that if $z_{ti} = 1$, then $\delta_i \geq \phi_{ti}$, $\delta_i \geq -\phi_{ti}$. Note also that the formulation readily adapts to the mean squared error, by changing the objective function to $n_{\text{samp}}^{-1} \sum_{i=1}^{n_{\text{samp}}} \delta_i^2$.

Combining these elements yields the affine hyperplane formulation,

$$\min \frac{1}{n_{\text{samp}}} \sum_{i=1}^{n_{\text{samp}}} \delta_i$$

$$
\begin{aligned}
\text{s.t.} \quad & \delta_i \geq \phi_{ti} - (1 - z_{ti})M && \forall i \in [n_{\text{samp}}], \quad \forall t \in \mathcal{T}_L \\
& \delta_i \geq -\phi_{ti} - (1 - z_{ti})M && \forall i \in [n_{\text{samp}}], \quad \forall t \in \mathcal{T}_L \\
& \phi_{ti} = y_i - \text{poly}(x_i; c_t) && \forall i \in [n_{\text{samp}}], \quad \forall t \in \mathcal{T}_L \\
& a_m^\top x_i \geq b_m - 2(1 - z_{ti}) && \forall i \in [n_{\text{samp}}], \quad \forall t \in \mathcal{T}_L, \quad \forall m \in A_R(t) && \text{(12a)} \\
& a_m^\top x_i + \mu \leq b_m \\
& \qquad + (2 + \mu)(1 - z_{ti}) && \forall i \in [n_{\text{samp}}], \quad \forall t \in \mathcal{T}_L, \quad \forall m \in A_L(t) && \text{(12b)} \\
& \textstyle\sum_{t \in \mathcal{T}_L} z_{ti} = 1 && \forall i \in [n_{\text{samp}}] && \text{(12c)} \\
& z_{ti} \leq l_t && \forall i \in [n_{\text{samp}}], \quad \forall t \in \mathcal{T}_L && \text{(12d)} \\
& \textstyle\sum_{i=1}^{n_{\text{samp}}} z_{ti} \geq N_{\min} l_t && \forall t \in \mathcal{T}_L && \text{(12e)} \\
& \textstyle\sum_{j=1}^{d} (a_{mj}^+ + a_{mj}^-) = 1 && \forall m \in \mathcal{T}_B \\
& a_{mj} = a_{mj}^+ - a_{mj}^- && \forall j \in [n], \quad \forall m \in \mathcal{T}_B \\
& a_{mj}^+ \leq o_{mj} && \forall j \in [n], \quad \forall m \in \mathcal{T}_B \\
& a_{mj}^- \leq (1 - o_{mj}) && \forall j \in [n], \quad \forall m \in \mathcal{T}_B \\
& z_{ti}, l_t \in \{0, 1\} && \forall i \in [n_{\text{samp}}], \quad \forall t \in \mathcal{T}_L \\
& o_{mj} \in \{0, 1\} && \forall j \in [n], \quad \forall m \in \mathcal{T}_B \\
& a_{mj}, b_m \in [-1, 1] && \forall j \in [n], \quad \forall m \in \mathcal{T}_B \\
& a_{mj}^+, a_{mj}^- \in [0, 1] && \forall j \in [n], \quad \forall m \in \mathcal{T}_B \\
& \phi_{ti} \in \mathbb{R} && \forall i \in [n_{\text{samp}}], \quad \forall t \in \mathcal{T}_L
\end{aligned}
$$

The constraints Eqs. (12a)–(12e) are the optimal classification tree with hyperplanes (OCT-H) of Bertsimas & Dunn (2017); the other constraints are our extension thereof.

*Remark* 3 (Axis-aligned regression).  In Appendix C, we propose a version of (12) tailored to functions whose full-dimensional strata have boundaries aligned with cartesian axes. Such functions appear in signal processing applications, and are the topic of recent works Chatterjee & Goswami (2021); Bertsimas & Dunn (2017); Madrid Padilla et al. (2021).

## C    Axis-aligned regression formulation

In this section, we introduce a variant of the affine hyperplane regression tree, presented in Appendix B, that accommodates using hyperplanes aligned with cartesian for partitioning the space. Tables 3 and 4 summarize the hyperparameters and variables of the mixed-integer formulation.

At each branching node $m \in \mathcal{T}_B$, a hyperplane splits the space in two subspaces

$$a_m^\top x_i - b_m < 0 \qquad a_m^\top x_i - b_m \geq 0, \tag{13}$$

that will be associated to the left and right children of node $m$. The parameters $a_m \in \{0,1\}^n$ and $b_m \in [0,1]$ are variables of the mixed integer program. In the axis-aligned formulation, the hyperplane is aligned with an axis of the cartesian space. This is enforced by requiring for all branching node $m \in \mathcal{T}_B$ that $a_m$ take boolean values, only one of which is one:

$$\sum_{j=1}^{d} a_{mj} = 1, \qquad 0 \leq b_m \leq 1.$$

Since $x_i \in [0,1]$, there holds $a_m^\top x_i \in [0,1]$, so that $b_m$ is constrained to $[0,1]$ without loss of generality.

In order to model the strict inequality in (13), we follow Bertsimas & Dunn (2017) and introduce the vector $\varepsilon \in \mathbb{R}^n$ of smallest increments between two distinct consecutive values in points $(x_i)_{i=[n_{\mathrm{samp}}]}$ in any dimension:

$$\varepsilon_j = \min\left\{ x_j^{(i+1)} - x_j^{(i)}, \text{ for } i \in [n-1] \; : \; x_j^{(i+1)} \neq x_j^{(i)} \right\}$$

where $x_j^{(i)}$ is the $i$-th largest value in the $j$-th dimension. $\varepsilon_{\max}$ is the highest value of $\varepsilon_j$ and serves as a tight big-M bound, leading to the formulation

$$
\begin{aligned}
a_m^\top x_i &\geq b_m - (1 - z_{ti}) & \forall m \in A_R(t) \\
a_m^\top (x_i + \varepsilon) &\leq b_m + (1 + \varepsilon_{\max})(1 - z_{ti}) & \forall m \in A_L(t)
\end{aligned}
$$

both for all $i$ in $[n_{\mathrm{samp}}]$ and all $t$ in $\mathcal{T}_L$. Recall that $z_{ti}$ takes binary values and is equal to one if sample $x_i$ belongs to leaf node $t$.

Combining these elements yields the axis-aligned formulation:

$$
\begin{aligned}
\min \quad & \frac{1}{n_{\mathrm{samp}}} \sum_{i=1}^{n_{\mathrm{samp}}} \delta_i \\
\text{s.t.} \quad & \delta_i \geq \phi_{ti} - (1 - z_{ti})M & \forall i \in [n_{\mathrm{samp}}], \quad \forall t \in \mathcal{T}_L \\
& \delta_i \geq -\phi_{ti} - (1 - z_{ti})M & \forall i \in [n_{\mathrm{samp}}], \quad \forall t \in \mathcal{T}_L \\
& \phi_{ti} = y_i - \mathrm{poly}(x_i; c_t) & \forall i \in [n_{\mathrm{samp}}], \quad \forall t \in \mathcal{T}_L \\
& a_m^\top x_i \geq b_m - (1 - z_{ti}) & \forall i \in [n_{\mathrm{samp}}], \quad \forall t \in \mathcal{T}_L, \quad \forall m \in A_R(t) & (14\text{a}) \\
& a_m^\top (x_i + \varepsilon) \leq b_m \\
& \qquad + (1 + \varepsilon_{\max})(1 - z_{ti}) & \forall i \in [n_{\mathrm{samp}}], \quad \forall t \in \mathcal{T}_L, \quad \forall m \in A_L(t) & (14\text{b}) \\
& \sum_{t \in \mathcal{T}_L} z_{ti} = 1 & \forall i \in [n_{\mathrm{samp}}] & (14\text{c}) \\
& z_{ti} \leq l_t & \forall i \in [n_{\mathrm{samp}}], \quad \forall t \in \mathcal{T}_L & (14\text{d}) \\
& \sum_{i=1}^{n_{\mathrm{samp}}} z_{ti} \geq N_{\min} l_t & \forall t \in \mathcal{T}_L & (14\text{e}) \\
& \sum_{j=1}^{d} a_{mj} = 1 & \forall m \in \mathcal{T}_B & (14\text{f}) \\
& 0 \leq b_m \leq 1 & \forall m \in \mathcal{T}_B & (14\text{g}) \\
& z_{ti}, l_t \in \{0,1\} & \forall i \in [n_{\mathrm{samp}}], \quad \forall t \in \mathcal{T}_L & (14\text{h}) \\
& a_{mj} \in \{0,1\} & \forall j \in [n], \quad \forall m \in \mathcal{T}_B & (14\text{i}) \\
& \phi_{ti} \in \mathbb{R} & \forall i \in [n_{\mathrm{samp}}], \quad \forall t \in \mathcal{T}_L
\end{aligned}
$$

The constraints Eqs. (14a)–(14i) are the optimal classification trees (OCT) of Bertsimas & Dunn (2017), whereas the other constraints are our extensions thereof. Note that we do not use the complexity parameters of the OCT formulation ($d_t$) and replace them with 1, where appropriate.

## D  COMPLEMENTARY EXPERIMENTAL RESULTS

In this section, we give complementary experiments that illustrate the practical behavior of the axis-aligned and affine-hyperplane regression models. The setup is identical to the one described in Section 4.2. We consider three additional regression problems, for which we plot the landscapes in the forthcoming figures and give approximation errors in Table 6.

### D.1 PIECEWISE-LINEAR NORMS

We consider two simple piecewise linear test functions: the $\| \cdot \|_1$ and $\| \cdot \|_\infty$ norms, defined for $x \in \mathbb{R}^n$ by

$$\|x\|_1 = \sum_{i=1}^{n} |x_i|, \qquad \|x\|_\infty = \max_{i \in [n]} |x_i|.$$

For both functions, we set depth $D = 2$ and polynomial degree $N = 1$. We sample $n_{\text{samp}} = 250$ points in the approximation space. We test both the axis-aligned (14) and the general affine-hyperplane (12) formulations with a time limit of 5 minutes for each optimization. For both norms, the axis-aligned formulation was solved to optimality and the affine-hyperplane formulation timed out.

Figure 8a shows the results on the $\| \cdot \|_1$ norm. Note that the full-dimensional strata of the $\| \cdot \|_1$ are axis aligned, so the axis-aligned formulation (14) (with $D = 2$) recovers both the correct strata and the correct polynomial function on each piece. The more general affine-hyperplane formulation (12) with performs equally well.

Figure 8b shows the results for the $\| \cdot \|_\infty$. The axis-aligned formulation (14) with depth yields a piecewise polynomial function that performs poorly at approximating the function. This is reasonable, as the full-dimensional strata are not axis-aligned anymore. The affine-hyperplane formulation (12) yields a piecewise polynomial function that matches the strata of the function, as well as the polynomial expression of the function on the strata.

Numerical results for both norms are in Table 6. They show a slight increase in error when using the affine-hyperplane formulation on the $\| \cdot \|_1$ norm. This can be attributed to the fact that the true partitioning is axis-aligned, which agrees with the main limitation of the axis-aligned formulation. And because the axis-aligned formulation is simpler to optimize, we obtain a provably optimal solution. Despite that, the solution of affine-hyperplane formulation has only slightly worse error. Additionally, looking at the errors for the $\| \cdot \|_\infty$ norm, we see order(s) of magnitude improvements in the error, when using the affine-hyperplane formulation. This underlines the increased expressiveness of the more general formulation.

### D.2 AN ADDITIONAL NEURAL NETWORK APPROXIMATION

Similarly to Section 4.2, we consider a similar NN with 25 parameters (2 hidden layers with 4 and 2 neurons respectively), but with *only ReLU* activation functions. The network is trained to minimize the mean squared error with the 2-dimensional function $x \mapsto 2\sin x_1 + 2\cos x_2 + x_2/2$ taken at 10 random points from the input space. All data is processed in a single batch, for 5000 epochs, using `AdamW` optimizer.

The piecewise linear approximation is obtained from the affine-hyperplane formulation with depth $D = 2$ and $n_{\text{samp}} = 225 = 15^2$ sample points (shown as red crosses) placed on a regular grid of $15 \times 15$ points. The degree of the polynomial pieces is $N = 1$. The MIP optimization times out after 5 minutes.

Figure 9a shows the landscape of the network (left pane) and the piecewise linear approximation (right pane). The obtained piecewise polynomial function essentially recovers the stratification of the network. Figure 9b presents the difference between the NN output and the approximation. The approximation recovers the slope of the network correctly on each of the full-dimensional strata. The discrepancy between the two functions is caused by the slight mismatch between the stratification of the network and its approximation, which could arguably be improved by taking samples from a denser grid.

Looking at the last row of Table 6, we notice that the median error is the lowest among all functions by orders of magnitude, pointing to the high quality of the fit of the polynomials in each partition. This might be due to the properties of taking points on a regular grid which might allow for better approximation.

### D.3 EMPIRICAL DEMONSTRATION OF THE APPROXIMATION ERROR

When approximating the Neural Network (Figure 5), we also perform a grid evaluation for varying tree depth $D \in \{1, 2, 3, 4\}$ and polynomial degree $N \in \{1, 2, 3, 4, 5\}$ to showcase the approximation

**Table 2:** Normalized absolute error for the denoising scenarios in Figure 12. The error is computed as the absolute difference between the ground truth signal and its approximation by the *axis-aligned* trees, divided by the maximal value of the ground truth.

| Denoising | max. err. | mean err. | median err. |
|---|---|---|---|
| Scenario 1 | 1.1 | $1.4 \times 10^{-1}$ | $9.3 \times 10^{-2}$ |
| Scenario 2 | 1.5 | $1.8 \times 10^{-1}$ | $9.0 \times 10^{-2}$ |
| Scenario 3 | $9.5 \times 10^{-1}$ | $1.2 \times 10^{-1}$ | $5.5 \times 10^{-2}$ |
| Scenario 4 | $8.3 \times 10^{-1}$ | $4.1 \times 10^{-2}$ | $1.6 \times 10^{-2}$ |

**Table 3:** Summary of the hyperparameters

| parameter | interpretation |
|---|---|
| $D$ | depth of the binary tree |
| $N_{min}$ | minimal number of points allowed in a nonempty leaf |
| $N$ | maximum degree of the polynomial on each piece |

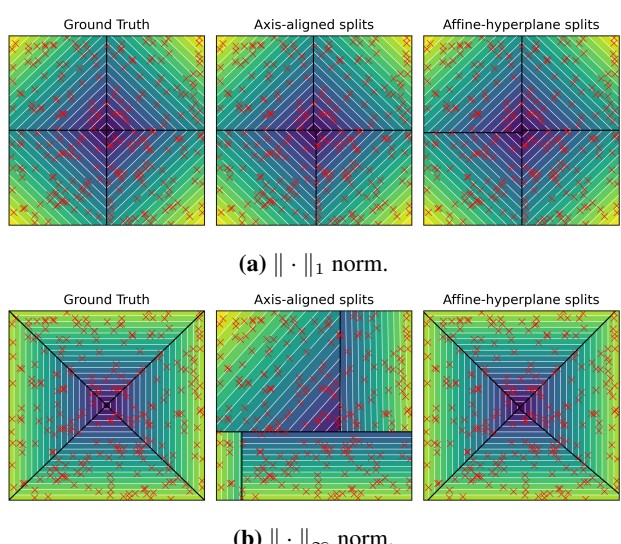

**(a)** $\| \cdot \|_1$ norm.

**(b)** $\| \cdot \|_\infty$ norm.

**Figure 8:** Approximation of $\ell_1$ and $\ell_\infty$ norms: original function (left), the axis-aligned approximation (14) (middle), and the (more general) affine-hyperplane formulation (12) (right). Red crosses indicate the samples used in the optimization. White lines show the level lines of the function; black lines indicate the boundaries of the full-dimensional strata.

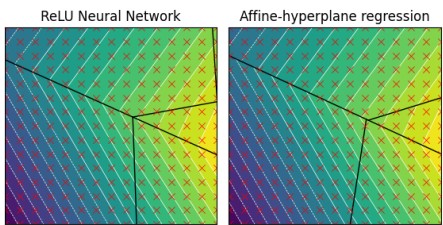

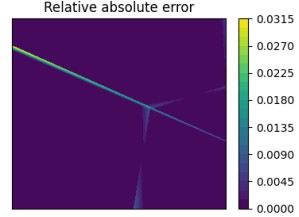

**(a)** Neural Network landscape (left) and its approximation (right). The black lines show the partitioning of the space, white lines are level lines, and red crosses show coordinates of sampled points.

**(b)** Normalized absolute error of the ReLU NN approximation by the tree model.

**Figure 9:** Approximation of the Neural Network with only ReLU activations. The errors are divided by the maximum absolute value of the NN output. Figure 9b shows that the approximation error is mostly due to slight inaccuracies in the partitioning; see Table 6 for quantitative errors.

**Table 4:** Summary of the variables of the affine-hyperplane regression tree formulation

| variable | index domain | interpretation |
|---|---|---|
| $l_t \in \{0,1\}$ | $t \in \mathcal{T}_L$ | 1 iff any point is assigned to leaf $t$ |
| $z_{ti} \in \{0,1\}$ | $t \in \mathcal{T}_L, i \in [n_{\text{samp}}]$ | 1 iff point $x_i$ is assigned to leaf $t$ |
| $a_m \in \mathbb{R}^n$ | $m \in \mathcal{T}_B$ | coefficients of the affine cut |
| $b_m \in \mathbb{R}$ | | |
| $o_{mj} \in \{0,1\}$ | $m \in \mathcal{T}_B, j \in [n]$ | 1 iff coordinate $j$ of $a_m$ is positive |
| $a_m^+, a_m^- \in \mathbb{R}$ | $m \in \mathcal{T}_B$ | the positive and negative part of $a_m$ |
| $\phi_{ti} \in \mathbb{R}$ | $t \in \mathcal{T}_L, i \in [n_{\text{samp}}]$ | fit error of point $x_i$ by the polynomial of leaf $t$ |
| $\delta_i \in \mathbb{R}$ | $i \in [n_{\text{samp}}]$ | fit error of point $x_i$ by the piecewise polynomial function |
| $c_t \in \mathbb{R}^{\binom{N+n-1}{n-1}}$ | $t \in \mathcal{T}_L$ | coefficients of the degree $N$ polynomial associated with leaf $t$ |

**Table 5:** Summary of the variables of the axis-aligned regression tree formulation

| variable | index domain | interpretation |
|---|---|---|
| $l_t \in \{0,1\}$ | $t \in \mathcal{T}_L$ | 1 iff any point is assigned to leaf $t$ |
| $z_{ti} \in \{0,1\}$ | $t \in \mathcal{T}_L, i \in [n_{\text{samp}}]$ | 1 iff point $x_i$ is assigned to leaf $t$ |
| $a_m \in \{0,1\}^n$ | $m \in \mathcal{T}_B$ | coefficients of the axis-aligned cut |
| $b_m \in \mathbb{R}$ | | |
| $\phi_{ti} \in \mathbb{R}$ | $t \in \mathcal{T}_L, i \in [n_{\text{samp}}]$ | fit error of point $x_i$ by the polynomial of leaf $t$ |
| $\delta_i \in \mathbb{R}$ | $i \in [n_{\text{samp}}]$ | fit error of point $x_i$ by the piecewise polynomial function |
| $c_t \in \mathbb{R}^{\binom{N+n}{n}}$ | $t \in \mathcal{T}_L$ | coefficients of the degree $N$ polynomial associated with leaf $t$ |

error bound in practice. The setup is otherwise the same as in the main experiments, except that the solver is run for 20 hours and the memory is increased to 64GB to ensure that even the largest run fits.

Figure 10 displays the results. Note that only the runs with depth $D = 1$ were solved to optimality. The error improves with increasing tree depth. The error also improves with increasing the polynomial degree from 1 to 2, but not so clearly for the higher degrees. This can be caused either by overfitting to the training data or by the optimization being more complicated, leading to a higher MIP gap, i.e., solutions with (potentially) worse quality at the time limit.

### D.4 SCALABILITY

Importantly, direct Mixed-Integer Programming approaches often struggle with scalability. This is also the case for the proposed method. While MIP solvers are becoming notably more capable (Koch et al., 2022), they do not scale well to functions with high dimensions. Note, however, that the formulation is independent of the size of the Neural Network model representing a given function. For higher-dimensional functions, one can use alternative methods of learning regression trees, such as Tree Alternating Optimization (TAO) (Carreira-Perpinan & Tavallali, 2018), to obtain well-performing trees without optimality guarantees.

To showcase the scalability of this direct approach, we report the results of approximating a 10-dimensional tame function, specifically a neural network with three hidden layers of sizes [20, 20, 10] and activation functions [ReLU, sigmoid, tanh]. The NN is trained on 10,000 samples from the multidimensional generalized Rosenbrock function. The setup is otherwise similar to the approximation results from Section D.3, we report results over a grid of tree depths and polynomial degrees. Each tree is trained for 20 hours using 10,000 uniformly distributed samples.

Figure 11 shows the absolute errors of each configuration. The scale of the error plays a bigger role in this evaluation, as can be seen from the overall improvement in the absolute relative error with increasing depth, which is not as clear from the absolute error values. Not even the depth-1 trees were solved to optimality in the 20-hour time span, and memory had to be increased to 256GB to ensure that the solving process fits into memory.

**Table 6:** Normalized absolute error between the functions and their approximations by the axis-aligned (14) and affine-hyperplane trees (12) of depth 2. The error is computed on a $1000 \times 1000$ grid of regularly spaced points. We divide the absolute errors by the maximal absolute value of the underlying ground truth to improve comparability.

| Test function | Axis-aligned tree (14) | | | Affine-hyperplane tree (12) | | |
|---|---|---|---|---|---|---|
| | max. err. | mean err. | median err. | max. err. | mean err. | median err. |
| $\|\cdot\|_1$ | $2.2 \times 10^{-2}$ | $9.8 \times 10^{-5}$ | $3.0 \times 10^{-5}$ | $2.6 \times 10^{-2}$ | $1.1 \times 10^{-4}$ | $3.0 \times 10^{-5}$ |
| $\|\cdot\|_\infty$ | $7.2 \times 10^{-1}$ | $8.5 \times 10^{-2}$ | $2.8 \times 10^{-2}$ | $6.6 \times 10^{-2}$ | $4.2 \times 10^{-4}$ | $5.0 \times 10^{-5}$ |
| ReLU NN | - | - | - | $3.0 \times 10^{-2}$ | $1.8 \times 10^{-4}$ | $2.7 \times 10^{-7}$ |

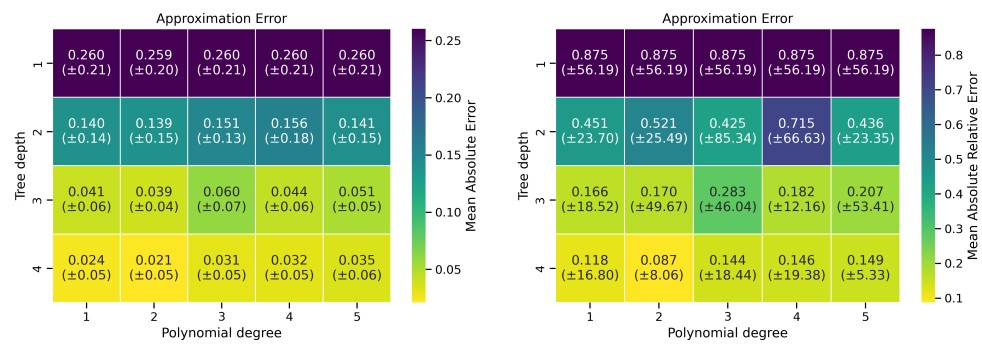

**(a)** Absolute error

**(b)** Absolute error, relative to the approximated value.

**Figure 10:** Mean absolute error, computed on a $1000 \times 1000$ point grid of a neural network approximation, similar to the configuration in the main text. Standard deviation over the data is reported in parentheses.

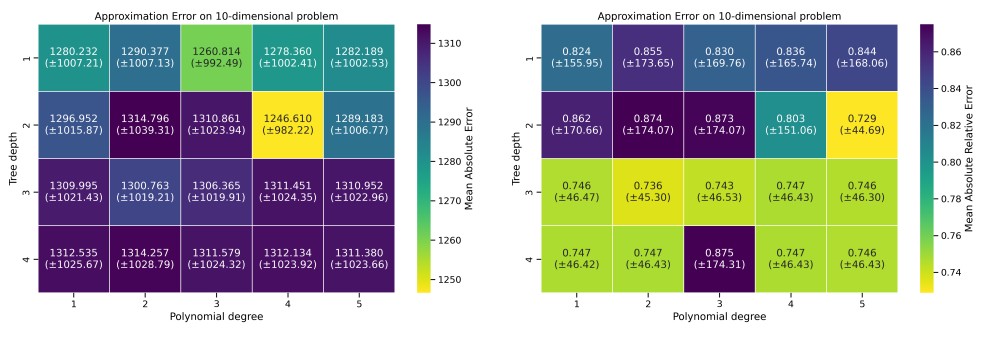

**(a)** Absolute error

**(b)** Absolute error, relative to the approximated value.

**Figure 11:** Results of approximating a neural network with a 10-dimensional input. Absolute error is computed on 1,000,000 uniformly sampled points from the 10-dimensional input space. We report mean and standard deviation (in parentheses).

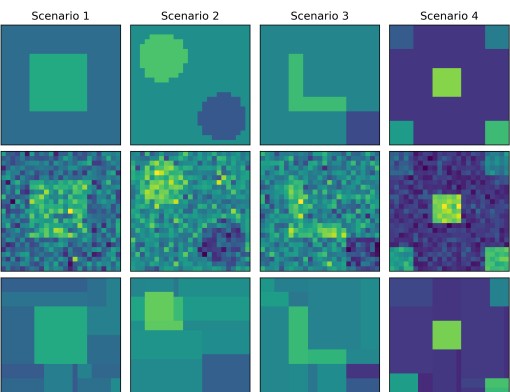

**Figure 12:** Four denoising scenarios from Madrid Padilla et al. (2021, Figure 3). The regression trees are restricted to axis-aligned splits and have depth 4. First row: ground truth, second row: (corrupted) regression signal, third row: recovered signal from the mixed integer formulation.

### D.5 DENOISING OF A PIECEWISE-CONSTANT 2D SIGNAL

We finally consider a slightly different set-up: following Madrid Padilla et al. (2021), we consider four two-dimensional functions that are piecewise constant on a 25 by 25 grid, illustrated in Figure 12 (row one). The regression data is the function corrupted by an additive Gaussian with zero mean and standard deviation of $\sigma = 0.5$, illustrated in Figure 12 (row two). Row three of Figure 12 shows, for each scenario, the output of regression trees with depth $D = 4$ and polynomial degree $N = 0$, with a time limit of 1 hour.

The piecewise polynomial formulation used here is a simplified version of Eq. (12), where the splits are restricted to align with the Cartesian axes; see Appendix C for details. Table 2 shows the recovery error of the regression trees; the recovered signals are comparable in quality to Figure 3 in Madrid Padilla et al. (2021).

