# OpenReview forum: "Piecewise Polynomial Regression of Tame Functions via Integer Programming"
_ICLR.cc/2026/Conference — Submitted to ICLR 2026_

### Official Review · Reviewer_A4b4 · 2025-10-31

**Soundness:** 3
**Presentation:** 2
**Contribution:** 2
**Rating:** 4
**Confidence:** 4

**Summary:**

The paper titled “PIECEWISE POLYNOMIAL REGRESSION OF TAME FUNCTIONS VIA INTEGER PROGRAMMING” presents a method for piecewise polynomial regression of tame functions, which is  a class of non-smooth, non-convex functions. This is done by first showing bounds on the decomposition of a non-convex function into piecewise polynomials and then formulating the regression problem as a Mixed-Integer Program. The core contributions include the first theoretical bound on the approximation error for generic tame functions, showing that the error decreases as the polynomial degree and the number of partition pieces (controlled by tree depth) increase. The practical side of this approach is demonstrated by showing effective approximation of functions like a mixed-activation neural network.

**Strengths:**

This paper claims to show the first theoretical bound on the approximation error of generic nonsmooth and nonconvex tame functions by piecewise polynomial functions.

Overall it gives a way to compute the best piecewise polynomial function, where each piece is a polyhedron defined by a number of affine inequalities, and the polynomial on each piece has a given degree.

**Weaknesses:**

While the paper’s experiments approximates  a small neural network example, the MIP approach's scalability is limited to low-dimensional functions in terms of proving global optimality. For typical high-dimensional models, the full optimization process will be prohibitively slow, as it is well known that MIPs takes hours to solve. This suggests the method is not practically viable for finding globally optimal solutions for large, modern neural networks.


The paper's primary theoretical contribution is a general approximation bound for tame functions and its core practical solution is a Mixed-Integer Programming (MIP) formulation. Given the focus on o-minimal structures,and  stratification theorems, the paper is arguably a better fit for an optimization or theoretical mathematics conference than a major DL conference like ICLR, where the scalability of solving the underlying optimization problem is a crucial metric for practical relevance.

What is not answered explicitly is what value the approximation adds from a practical point of view.
It hints at several potential applications but any solid experimental results backing it up is lacking.
This makes the paper and its proposed approach, more of a proof-of-concept for tame functions rather than a practical tool for the deep learning community

**Questions:**

na

---

> ### Author Response · Authors · 2025-12-03
>
> We thank the reviewer for their thoughtful feedback. Although the discussion is, at this point, essentially over, we include our intended answer for reference.
>
> Here are some point-by-point answers. The changes in the paper appear in blue.
>
> > a better fit for an optimization or theoretical mathematics conference than a major DL conference
>
> We would argue that the theory of deep learning has a long-running tradition in major DL conferences. Compared to previous papers in the approximation end of DL theory, such as:
> https://proceedings.mlr.press/v38/balazs15.html
> which focused on Lipschitz functions, we focus on a class of locally Lipschitz functions that includes ReLU. This makes the results directly relevant to DL practitioners.
>
> > While the paper’s experiments approximates a small neural network example, the MIP approach's scalability is limited to low-dimensional functions in terms of proving global optimality.
>
> True, the MIP gets very difficult to solve as the dimension increases. However, our main point in this paper is that most DNNs (thanks to the tame assumption) are approximable by the piecewise polynomial functions, and that the proposed formulation is one way to construct such approximations. The way to solve the formulation is beyond our contribution: users who insist on global optimality certificates should use solvers such as Cplex or Gurobi; otherwise, many approaches for solving the MIP are applicable, such as first-order methods or TAO [1].
>
> The affine-tree formulation (page 17) grows linearly in data size and leaf count (both in terms of constraints and variables). For the dimension and the polynomial degree, the number of coefficient variables grows exponentially, while the number of constraints does not change with increasing the degree. In general, solving MIO formulations is exponential in the number of variables and constraints. Empirically, however, we show that we obtain high-quality solutions in little time (see Figure 6). And there is substantial empirical evidence for the increase in MIP solver speed across a diverse benchmark in the last 20 years [2]. Extrapolating this progress, future solvers might make the same formulation scale to larger problems.
>
> References:
>
> [1] Alternating optimization of decision trees, with application to learning sparse oblique trees, (Neurips 2018) M. A. Carreira-Perpinan, P. Tavallali
>
> [2] Progress in mathematical programming solvers from 2001 to 2020, (EURO JCO 2022) Thorsten Koch, Timo Berthold, Jaap Pedersen, Charlie Vanaret.

---

### Official Review · Reviewer_W6es · 2025-10-31

**Soundness:** 2
**Presentation:** 3
**Contribution:** 2
**Rating:** 4
**Confidence:** 3

**Summary:**

The paper studies approximation of tame functions using piecewise polynomial models. It formulates an MIP approach that learns an affine-hyperplane tree to partition the input space and fits a polynomial on each leaf, implemented via monomial lifting and standard linearizations.

**Strengths:**

The main innovations are the tidy, two-term error characterization for general tame functions and a practically solvable MIP formulation that operationalizes piecewise polynomial regression beyond axis-aligned trees.

**Weaknesses:**

However, I have the following concerns:

1. The problem framing is tailored to be theory-friendly. The tame assumption is difficult to verify and can be fragile under noise or distribution shift, so it is unclear whether the structural guarantees meaningfully carry over to practical implementations. Besides, the proposed approach relies on monomial lifting and discrete tree decisions, which causes rapid growth in variable counts and branching complexity. The experiments remain small-scale and do not convincingly demonstrate that the method remains tractable as dimensionality or dataset size increases. Finally, the work offers limited guidance on stopping criteria, numerical stability, or deployment playbooks.

2. For medium-to-high dimensional regression or large datasets, the training and maintenance overhead of a MIP-based piecewise polynomial model will generally be prohibitive. Tasks with highly intricate non-smooth boundaries will force deeper trees and many leaves, further increasing the combinatorial burden. In strongly real-time applications that must update models continually, mainstream methods such as gradient-boosted decision trees, random forests, neural networks, kernel methods, and spline/FEM surrogates tend to reach useful accuracy much faster and with lower engineering cost.

3. As I know, there is no automated, provably stable strategy for setting and tightening big-M constants or for adapting numeric scales, which are essential to avoid weak relaxations and solver pathologies. The model complexity is not adapted to choose polynomial degree or leaf capacity based on error indicators, so accuracy improvements inevitably come with combinatorial blow-ups.

4. When evaluated on the axes of error versus training time versus memory, the proposed approach is unlikely to be superior to GBDT/LightGBM, modern neural regressors, or spline/FEM surrogates at moderate or large scales. The method’s edge is interpretability and constraintability, since leafwise polynomials can be audited and global constraints such as monotonicity or safety can be encoded, but the paper does not quantify this advantage against strong baselines that already support monotonic constraints, fairness regularizers, or distillation for interpretability.

**Questions:**

1. How to diagnose whether the tame structure is a reasonable working assumption on noisy, shifting datasets？

2. What are the empirical and analytical scaling laws linking dimension, data size, polynomial degree, and leaf count to time, memory, and solver gap？

---

> ### Author Response · Authors · 2025-11-27
>
> We thank the reviewer for their thoughtful feedback, and apologize for the delay in answer.
> We are pleased that the reviewer appreciated our two-term error characterization for general tame functions, and the practically solvable MIP formulation valid beyond axis-aligned trees.
>
> Here are some point-by-point answers. The changes in the paper appear in blue.
>
> > W1. The problem framing is tailored to be theory-friendly. The tame assumption is difficult to verify and can be fragile under noise or distribution shift, so it is unclear whether the structural guarantees meaningfully carry over to practical implementations.
> > Q1. How to diagnose whether the tame structure is a reasonable working assumption on noisy, shifting datasets？
>
>
> On this point, we strongly disagree. The tame assumption is very easy to verify. Indeed, since it is stable by composition, addition, multiplication, and inverse, among others, it is enough to check that all the elementary constituents of the DNN are tame. Tame functions include any polynomial function, including linear, lp norm, exponential, logarithm, and sigmoid. It is therefore easy to check that a DNN is tame; we refer to [2] and references therein for a precise account showing that $96\%$ out of $400$ activation functions are tame. The tameness of the optimization problem is fully captured by the functions it features, and is independent of the data points and their potential shifts.
>
> > (W1 cont.) Besides, the proposed approach relies on monomial lifting and discrete tree decisions, which causes rapid growth in variable counts and branching complexity. The experiments remain small-scale and do not convincingly demonstrate that the method remains tractable as dimensionality or dataset size increases.
>
> A major challenge in interpretability and approximation of deep neural networks, among other objects, is to propose methods that scale with dimension or dataset size. We do not claim that our formulation scales well, but then, no other method that we know about does scale well. See below for a discussion on the more specific issue of solving the MIP.
>
> > (W1 cont.) Finally, the work offers limited guidance on stopping criteria, numerical stability, or deployment playbooks.
>
> Our contribution lies in the theoretical analysis, which shows that any tame function can be arbitrarily approximated by piecewise polynomial functions, and in the MIP formulation (eq. 12). We feel that specifying which MIP solver and what parameters to employ to solve a given MIP is a benchmarking effort, rather distinct from the theoretical approach taken here. Nevertheless, a good first approach lies in using modern MIP solvers, either commercial (CPLEX, Xpress, Gurobi) or open-source (CBC, SCIP, HiGHS, ...). Besides, as the MIO problem features a decision tree, specialized MIO solvers that scale better, such as [1], may be applicable.
>
> > W2. For medium-to-high dimensional regression or large datasets, the training and maintenance overhead of a MIP-based piecewise polynomial model will generally be prohibitive. Tasks with highly intricate non-smooth boundaries will force deeper trees and many leaves, further increasing the combinatorial burden. In strongly real-time applications that must update models continually, mainstream methods such as gradient-boosted decision trees, random forests, neural networks, kernel methods, and spline/FEM surrogates tend to reach useful accuracy much faster and with lower engineering cost.
>
> True, the MIP gets very difficult to solve as the dimension increases. However, our main point in this paper is that most DNNs (thanks to the tame assumption) are approximable by the piecewise polynomial functions, and that the proposed formulation is one way to construct such approximations. The way to solve the formulation is beyond our contribution: users who insist on global optimality certificates should use solvers such as Cplex or Gurobi; otherwise, many approaches for solving the MIP are applicable, such as first-order methods or TAO [1].
>
> > W3. As I know, there is no automated, provably stable strategy for setting and tightening big-M constants or for adapting numeric scales, which are essential to avoid weak relaxations and solver pathologies. The model complexity is not adapted to choose polynomial degree or leaf capacity based on error indicators, so accuracy improvements inevitably come with combinatorial blow-ups.
>
> All major solvers, such as CPLEX or Gurobi, have the ability to tune the big\M constraints on the fly, tightening them as they see fit. More broadly, our contribution on the practical side lies in formulating the regression problem as a Mixed-Integer optimization problem; providing a recommendation on which solver to use is a benchmarking effort, outside the scope of the current paper.

---

> > ### Author Response · Authors · 2025-11-27
> >
> > > W4. When evaluated on the axes of error versus training time versus memory, the proposed approach is unlikely to be superior to GBDT/LightGBM, modern neural regressors, or spline/FEM surrogates at moderate or large scales. The method’s edge is interpretability and constraintability, since leafwise polynomials can be audited and global constraints such as monotonicity or safety can be encoded, but the paper does not quantify this advantage against strong baselines that already support monotonic constraints, fairness regularizers, or distillation for interpretability.
> >
> > We agree that, if our focus was about learning surrogate models, we should certainly discuss and compare to the mentioned methods. However, our main interest in that paper is to build approximations of tame functions.
> >
> > > Q2. What are the empirical and analytical scaling laws linking dimension, data size, polynomial degree, and leaf count to time, memory, and solver gap？
> >
> > The affine-tree formulation (page 17) grows linearly in data size and leaf count (both in terms of constraints and variables). For the dimension and the polynomial degree, the number of coefficient variables grows exponentially, while the number of constraints does not change with increasing the degree. In general, solving MIO formulations is exponential in the number of variables and constraints. Empirically, however, we show that we obtain high-quality solutions in little time (see Figure 6). And there is substantial empirical evidence for the increase in MIP solver speed across a diverse benchmark in the last 20 years [3]. Extrapolating this progress, future solvers might make the same formulation scale to larger problems.
> >
> >
> > References:
> >
> > [1] Alternating optimization of decision trees, with application to learning sparse oblique trees, (Neurips 2018) M. A. Carreira-Perpinan, P. Tavallali
> >
> > [2] Deep Learning as the Disciplined Construction of Tame Objects (2025)
> >
> > [3] Progress in mathematical programming solvers from 2001 to 2020, (EURO JCO 2022) Thorsten Koch, Timo Berthold, Jaap Pedersen, Charlie Vanaret.

---

### Official Review · Reviewer_6Gn6 · 2025-10-31

**Soundness:** 3
**Presentation:** 3
**Contribution:** 3
**Rating:** 4
**Confidence:** 4

**Summary:**

This paper develops a theory for the uniform approximation of tame functions by piecewise-polynomial functions. The proof makes use of the stratification of tame functions into smooth parts, combining the usual polynomial approximation theory of smooth functions with recently-developed techniques for estimating/controlling the singular set. The construction yields an algorithm for this approximation via mixed integer programming, which the authors demonstrate numerically on several simple test functions (norm, cone function, small neural network).

**Strengths:**

1. This paper is very clearly written, and the proof of the approximation theorem is easy to follow. I know it can sometimes make the math less clear when the theory has to be summarized in the main paper and proved in appendices, but I think this work does well at communicating the main technical ideas.

2. It is definitely valuable to understand the complexity of piecewise-polynomial approximation for broad classes of functions. I am unfamiliar with definable structures and such, but the piecewise-smooth structure of tame functions makes it a very natural class to analyze using this paper's argument (bootstrapping the smooth case using control on the strata). It is also sufficiently general to be interesting, as many functions of approximation interest are tame.

**Weaknesses:**

1. One thing that could be more clear is which elements of this paper are novel. (Every review of every paper in every ML conference says this so I will try to be specific). In my understanding, the approximation result combines classical polynomial approximation of smooth functions (Jackson bounds) with the results from [Boissonat et al, 2023] which control the singular set. The numerical approach uses standard polyhedral optimization machinery (decision trees, encoding as a MIP, etc.). These are definitely strong elements and the authors combine them naturally, but I do feel like the "related work/preliminaries" is a bit lacking in these departments. The authors emphasize that the stratification of tame functions is a pioneering classical result, but the other tools are not given the same exposition. I would like to know more about the fields/problems where they come from.

2. Having the constants in the approximation theorem depend on f is a scary thing... Of course your result is still saying something useful, but I think one thing about Jackson's bound (such as Theorem 1.7 of https://www.math.umd.edu/~petersd/666/amsc666notes02.pdf) that is crucial in application is that the exact dependence of the constant on the function f is known (it's just the ||\cdot||_{C^k} norm). I am not sure if the exact dependence of the constants on f and dimension can be identified (would be very helpful if so!), but I think it would be nice if the authors could say a bit about the behavior they expect. The main questions I have are (1) which types of singularities are more efficient to approximate than others and (2) how does the "cost" of singularities scale with dimension.

3. I think the experimental section is a bit underwhelming. The MIP approach does not seem efficient enough to scale to mid-size problems, and computational restrictions made it so that only very small regressions could be attempted. I know one of the reasons for excitement in piecewise-smooth approximation is due to modern neural net practices, but it's hard to gain insight from these experiments about larger-scale approximation questions. I am not sure how the authors want to position it, but it is tricky; the approximation result is strong and an optimization algorithm pairs nicely, but it seems improvements are needed on the optimization side.

**Questions:**

1. Naturally, I have the question corresponding to Weakness (1) above. Since I am not familiar with tame stuff, I can contextualize my question with an analogous situation in geometric measure theory (GMT). In this field, the notion of "rectifiable sets" was crafted to represent "sets which are covered (almost everywhere) by a stratification into smooth pieces". I think any reasonable person would try to apply the smooth polynomial approximation case (Jackson bounds) on the smooth parts and glue them together across the singular parts, and so the name of the game becomes understanding the structure of the singular set well enough to conclude nontrivial things. Sometimes there is a smaller class of geometric objects, such as minimal surfaces, for which we can say more about their singular sets, which naturally would yield similar results about "how to glue smooth results together when given a patchwork of smooth results". Many proofs in nonsmooth geometry have to address this at some point.

Sorry for the rant, but I share the above anecdote to say that there is a whole continuum of different results about "piecewise polynomials approximate class X with rate Y" which may be proved through this high-level argument. The tools are out there, and sometimes the results  don't get written up and proven. In my eyes, the value of such a result comes from an interesting/general function class X or a rate Y which is somehow informative. In the minimal surface setting, people took the time to write hundreds of pages about it because the class of minimal surfaces is interesting enough. In short, I would like to know more about how approximation of tame functions is thought of/used more widely, as well as if there is anything interesting about the error rate you get. Is the dependence on \ell and n surprising/optimal, does it agree with or improve on someone else's rate for a related problem? Upon reading the paper now, it is unclear how the choice of X=tame functions and the rate ell^{-1/n} fit with expectations or comparable results.


2. What does the rate \ell^{-1/n} suggest? I know Boris Hanin showed that for e.g. random ReLU nets (piecewise linear), the typical number of regions is much smaller than expected (poly(dim) instead of exp(dim)); where do you think your bound incurs any pessimism, and how does this approximation rate relate to the curse of dimensionality?

3. Can you share a bit more about Assumption 1? It appears to be some nondegeneracy/transversality thing, and I guess I am wondering if it is a merely technical assumption or an important element that somehow makes the singular set covering more efficient and allows you to get the rate you did. As I mentioned earlier, while reading this my main question was "what extra things are they using other than being piecewise-smooth?". It would definitely help me out if the authors could detail a bit more how the assumptions of tameness and Assumption 1 constrain the singularities in a way that makes the approximation construction more efficient (or, if it doesn't make it more efficient, why they are needed at all).

---

> ### Author Response · Authors · 2025-11-27
>
> We thank the reviewer for their thoughtful and detailed feedback, and apologize for the delay in answer.
> We are pleased that the reviewer found that our paper contributes to a valuable research question, and has an assumption that is both natural for the analysis, and sufficiently general to be interesting.
> We are glad that the reviewer found that our paper is very well written, does well at communicating the main ideas, and has an easy-to-follow proof in the appendix.
>
> Here are some point-by-point answers. The changes in the paper appear in blue.
>
> > W2. Having the constants in the approximation theorem depend on f is a scary thing... Of course your result is still saying something useful, but I think one thing about Jackson's bound (such as Theorem 1.7 of https://www.math.umd.edu/~petersd/666/amsc666notes02.pdf) that is crucial in application is that the exact dependence of the constant on the function f is known (it's just the ||\cdot||_{C^k} norm). I am not sure if the exact dependence of the constants on f and dimension can be identified (would be very helpful if so!), but I think it would be nice if the authors could say a bit about the behavior they expect. The main questions I have are (1) which types of singularities are more efficient to approximate than others and (2) how does the "cost" of singularities scale with dimension.
>
> We have worked out the dependence in $f$, and we thank the reviewer for raising this very interesting point.
> It turns out that the dependence of the bound to $f$ in our result, Theorem 1, is similar to that of Jackson results: the $\mathcal{C}^m$ norm of the restriction of function to the set where it is $\mathcal{C}^m$ (the union of the full-dimensional strata) appears as a multiplicative constant factor in the bound. We have updated the main result to highlight this.
>
> With our current proof techniques, we do not see how to work out the dependence of the bound to the dimension of the space $n$ or the smoothness parameter of the stratification $m$.
> Indeed, we notably use Theorem 1 of (Fefferman), which provides a bound where the constant depends on the space dimension, and the degree of regularity of the function; this is shown in equation (2) of our proof.
> Nevertheless, we note that classic Jackson results usually also don't show the exact dependence of the bound to the degree of smoothness of the function and the space dimension.
>
> In terms of computational cost, we expect that functions with a higher number of $n-1$-dimensional strata, or strata that have a higher curvature will be more difficult to approximate. For that second aspect, the easy situation is when the strata are actually linear, in which case we could get exact recovery (with $\ell$ large enough).
>
> > W1. One thing that could be more clear is which elements of this paper are novel. In my understanding, the approximation result combines classical polynomial approximation of smooth functions (Jackson bounds) with the results from [Boissonat et al, 2023] which control the singular set. The numerical approach uses standard polyhedral optimization machinery (decision trees, encoding as a MIP, etc.). These are definitely strong elements and the authors combine them naturally, but I do feel like the "related work/preliminaries" is a bit lacking in these departments. The authors emphasize that the stratification of tame functions is a pioneering classical result, but the other tools are not given the same exposition. I would like to know more about the fields/problems where they come from.
>
> We thank the reviewer for this opportunity to clarify the positioning.
> We have updated the related works section to provide a proper account of the theoretical results that we use, and to detail from what literature they originate.

---

> ### Author Response · Authors · 2025-11-27
>
> > Q1. [...] In short, I would like to know more about how approximation of tame functions is thought of/used more widely, as well as if there is anything interesting about the error rate you get. Is the dependence on \ell and n surprising/optimal, does it agree with or improve on someone else's rate for a related problem? Upon reading the paper now, it is unclear how the choice of X=tame functions and the rate ell^{-1/n} fit with expectations or comparable results.
>
> We do believe that the tame assumption is interesting enough!
> The tame assumption (tame topology) orginates from logic theory. It is a relatively young subject (first papers in 1980s), and regarded as a good candidate of the "topologie modérée" envisioned by Grothedieck in his Esquisse d'un programme (1984). In more recent years, the tame assumption is gaining popularity in the ML field because (i) it is _realistic_ in that it covers most problems and neural network architectures (it is stable by most mathematical operations such as composition, and in the context of ML $96\%$ of $400$ activation functions are definable, see [3] and references therein), and (ii) it has allowed to prove relevant results for ML (first result of convergence of SGD on nonsmooth nonconvex function [1], analysis of the chain rule on nonsmooth functions [2]).
> However, this is not yet a standard popular assumption, and we are thus not aware of approximation results that make use of the tame assumption. That is the main motivation behind our results.
> Theorem 1 assumes that the function is tame and continuous (plus minor things), and is at the same time crucial to the results. Indeed, dropping it would broaden the range to all continuous functions, which are notoriously hard to approximate (a subset of these are 1-Lipschitz continuous functions, they are almost everywhere non-differentiable).
> Thus, Theorem 1 provides an approximation result that applies to a very braod class of nonsmooth nonconvex continuous functions (we are not aware of other approximation results in such a broad class).
> That being said, the bound itself aligns well with other approximation results: the first term $N^{-m}$ relates to the fast approximation rate brought by Jackson-type theorems, and the second $l^{-2/(n-1)}$ relates to the approximation of the nondifferentiable areas (the strata of dimension $n-1$) by piecewise linear functions.
>
> > Q2 What does the rate \ell^{-1/n} suggest? I know Boris Hanin showed that for e.g. random ReLU nets (piecewise linear), the typical number of regions is much smaller than expected (poly(dim) instead of exp(dim)); where do you think your bound incurs any pessimism, and how does this approximation rate relate to the curse of dimensionality?
>
> We restrict to approximation by piecewise polynomial functions with _linear_ boundaries, hence we incur the curse. With curved boundaries, we conjecture that we could get an improved bound. We did not pursue yet this direction as (i) in practice, working with nonlinear boundaries turns the linear mixed integer program into a nonlinear one, and (ii) the estimates of Boissonat only apply to piecewise linear approximation of manifolds, while we would need piecewise nonlinear (eg polynomial) approximation.
>
> One more surprising aspect is that the second term depends on the dimension of the space, and not the smoothness of boundary manifolds $m$ (provided by the stratification theorem).
>
> Finally, it is unclear to us how the results of Hanin could inform our work.
> Indeed, in the work cited in the related works, Hanin proposes to bound the number of linear regions by the number of neurons in the network.
> In our tame framework, that result could translate as bounding the number of full-dimensional strata by the number of "elementary & simple tame functions", in a sense that would need to be specified.
> However, we focus on how to approximate a given tame function by piecewise polynomial functions of given degree and number of pieces.

---

> ### Author Response · Authors · 2025-11-27
>
> > Q3 Can you share a bit more about Assumption 1? It appears to be some nondegeneracy/transversality thing, and I guess I am wondering if it is a merely technical assumption or an important element that somehow makes the singular set covering more efficient and allows you to get the rate you did. As I mentioned earlier, while reading this my main question was "what extra things are they using other than being piecewise-smooth?". It would definitely help me out if the authors could detail a bit more how the assumptions of tameness and Assumption 1 constrain the singularities in a way that makes the approximation construction more efficient (or, if it doesn't make it more efficient, why they are needed at all).
>
> The Assumption 1 is mainly of a technical nature, required to apply the results of Boissonat et al. Our conjecture is that a Verdier stratification can always be refined so that all the stratum meet Assumption 1.
> The impact would be to increase the cardinality (number of strata) of the new stratification, which would leave the bound unchanged.
> We did not pursue this proof as it would require to introduce many tools from logic and stratification theory that would be outside the scope of the current paper.
>
> More generally, and as discussed above, tameness is the crucial assumption here, in that it provides the piecewise smooth structure to the function, with finitely many smooth parts. Assumption 1 does not constrain the singularity of the function, merely the description of the singular sets in terms of the $\Phi$ mappings, that are used to call the results from Boissonat et al.
>
> > W3. I think the experimental section is a bit underwhelming. The MIP approach does not seem efficient enough to scale to mid-size problems, and computational restrictions made it so that only very small regressions could be attempted. I know one of the reasons for excitement in piecewise-smooth approximation is due to modern neural net practices, but it's hard to gain insight from these experiments about larger-scale approximation questions. I am not sure how the authors want to position it, but it is tricky; the approximation result is strong and an optimization algorithm pairs nicely, but it seems improvements are needed on the optimization side.
>
> True, the MIP gets very difficult to solve as the dimension increases. However, our main point in this paper is that most DNNs (thanks to the tame assumption) are approximable by the piecewise polynomial functions, and that the proposed formulation is one way to construct such approximations. The way to solve the formulation is beyond our contribution: users who insist on global optimality certificates should use solvers such as Cplex or Gurobi; otherwise, many approaches for solving the MIP are applicable, such as first-order methods or TAO [4].
>
> The affine-tree formulation (page 17) grows linearly in data size and leaf count (both in terms of constraints and variables). For the dimension and the polynomial degree, the number of coefficient variables grows exponentially, while the number of constraints does not change with increasing the degree. In general, solving MIO formulations is exponential in the number of variables and constraints. Empirically, however, we show that we obtain high-quality solutions in little time (see Figure 6). And there is substantial empirical evidence for the increase in MIP solver speed across a diverse benchmark in the last 20 years [5]. Extrapolating this progress, future solvers might make the same formulation scale to larger problems.
>
> References:
> [1] Jérôme Bolte and Edouard Pauwels. A mathematical model for automatic differentiation in machine learning. In Advances in Neural Information Processing Systems, volume 33, pp. 10809–10819. Curran Associates, Inc., 2020.
>
> [2] Damek Davis, Dmitriy Drusvyatskiy, Sham Kakade, and Jason D. Lee. Stochastic Subgradient Method Converges on Tame Functions. Foundations of Computational Mathematics, 20(1):119–154, February 2020. ISSN 1615-3383. doi: 10.1007/s10208-018-09409-5.
>
> [3] Gilles Bareilles, Allen Gehret, Johannes Aspman, Jana Lepšová, and Jakub Mareˇcek. Deep Learning
> as the Disciplined Construction of Tame Objects, September 2025. [https://arxiv.org/abs/2509.18025]
>
> [4] Alternating optimization of decision trees, with application to learning sparse oblique trees, (Neurips 2018) M. A. Carreira-Perpinan, P. Tavallali
>
> [5] Progress in mathematical programming solvers from 2001 to 2020, (EURO JCO 2022) Thorsten Koch, Timo Berthold, Jaap Pedersen, Charlie Vanaret.

---

> ### Comment · Reviewer_6Gn6 · 2025-11-28
> **Reply to Authors**
>
> I thank the authors for their time, consideration, discussion, and careful revisions (in the spirit of that, happy Thanksgiving everyone!). The identification of the dependence of the approximation constant via C^m norms is exactly what I was asking about, thank you. Furthermore, the linearity of the boundaries and the role it plays in your theory helps me place your result in my own intuition and understanding of approximation theory -- I see now that your rate is more or less what should be expected given the assumptions, and I quite like the positioning of this paper as groundwork and a logical first step toward a general approximation theory for tame functions. Curved boundaries are perhaps the next high-level step, and your response helps clarify which of your tools will generalize and which won't. Looking back at it now these ingredients were present in the original manuscript; certainly with more experience in tame functions these high-level questions (dependence on dimension/C^m smoothness, anticipated dependence on curvature of boundaries, etc.) would have been clearer. I still do think more emphasis could be placed on what we learn from these musings (which types of singularities are efficiently approximated across, how the definition of "singularities being efficiently approximated across" should be thought of w.r.t. dimension, and how to relate these ideas to other mathematical/geometric notions), and even though it is a loose hand-wavy question I think some exposition in this direction could be beneficial. That being said the authors have addressed many of my concerns, and I will raise my score accordingly :)
>
> Regarding optimization/MIP, I totally agree with the authors that MIP is simply an existence tool and in practice things will be approached with gradient descent. I ask about optimization experiments because I do think it is a good way to probe the question of "how hard is it for a reasonable algorithm to find this approximator". Of course, doing this carefully requires choosing one of a million architecture/approximation classes and making a billion arbitrary experimental decisions and definitely is out of scope for this paper. If the authors think there is some theory-based intuition or a simple experiment to run in this direction to produce some conjectures/vibes about how likely your approximants are to be found by the conventional pipeline (NN with a not-too-weird architecture and some type of gradient descent), I believe it would be a good boost for the community and open up directions for future research. At the moment you are probably the best people in the world to guess what architecture+optimizer is right for tame functions (or at least how conventional methods will behave), and I encourage the authors to open up the exploration into first-order optimization properties in the experimental section of this paper if they can find an easy way to do so. I will not count this side of the conversation toward my score -- it would be unfair since I have no idea how I would approach the issue (maybe by following up your "non-piecewise-linear neural network" example by training student NNs with e.g. polynomial activations from scratch and comparing with the MIP approximant?). All I can say is that as a fellow researcher I think there's a certain bravery in publishing a paper which asks questions before knowing the answer, and I always try to encourage it (especially when trying to reintroduce older mathematical ideas to the ML community).
>
> All in all, the authors' revisions addressed most of my technical questions, and my higher-level questions regarding positioning are answered well by the authors' responses (I think incorporating some of this stuff into the revised paper, particularly the discussion on linear vs curved boundary and how the linear case is good groundwork for the general case, would be helpful to readers). On the experimental side, the current experimental section is focused on verifying the theory in various applications (as it should be), but perhaps it could also benefit from some exploratory investigations into the questions on the deep learning practitioner's mind. I do not think there is need to blindly scale experiments up or put effort toward more intricate MIP solvers (since in my eyes the main result is the approximation bound)... if the authors have time I think it's more worthwhile to spend toward positioning/adding intuition/exploring implications. This paper is a solid first step in the interesting direction of "deep learning of tame functions", and to me the facts that (1) curved boundaries are avoided, (2) the relationship between neurons and strata remains blurry, and (3) first-order optimization properties of this approximant are unknown do not constitute weaknesses of this paper but rather a list of next steps to take.
>
> Lovely work
>
>
> edit: OpenReview will not let me increase my rating from 4 to 6 (or at all). This is sad :(

---

### Author Response · Authors · 2025-12-03
**Discussion Summary for AC**

Dear AC and reviewers,

We sincerely thank all reviewers and the AC for the time and care dedicated to evaluating our submission, and for the many constructive suggestions.

We are grateful for the support of Reviewer 6Gn6, who was willing to increase the score during the discussion. Reviewers W6es and A4b4 could not comment on our answers, and thus decide on assessment updates.

Our contributions are summarized below

1. **Pioneering approximation bound**: we provide the first bound on the approximation quality of any tame function by piecewise polynomial functions (Rev. 6Gn6, W6es, A4b4)
2. **Approximation by a Mixed Integer Program**: we provide a MIP forumalation that allows to compute or approximate the piecewise polynomial function obtained in the above bound (Rev. 6Gn6, W6es, A4b4)

During the discussion phase, we provided detailed clarifications and additional results, summarized here:

- **Easy to check and realistic tame assumption**: the tame assumption is very easy to verify, and covers most Deep Learning setups. Indeed, since it is stable by composition, addition, multiplication, and inverse, among others, it is enough to check that all the elementary constituents of the DNN are tame. Tame functions include any polynomial function, including linear, lp norm, exponential, logarithm, and sigmoid (Rev. 6Gn6, W6es)
- **Improved precision of main bound**: prompted by Rev. 6Gn6, we reviewed our proof and the revised pdf now includes explicitly the dependence of the bound to the function explicitly, a notable improvement (Rev. 6Gn6)
- **Improved exposition of methods**: we provide in the revised pdf more context for the approximating theory state-of-the-art, and the theoretical tools we use (Rev 6Gn6).
- **Clarified use of MIP**: the MIP is simply a middle ground between a purely existential result, "there exists a piece-wise polynomial approximation", and the practice, of using some first-order method (Rev. 6Gn6, W6es). We refer to the answer to Rev. W6es for details. We have also included more experiments on scalability in the revised Section D.4. Additionally, we empirically demonstrate the approximation bound in Section D.3.

We especially appreciate the consistent recognition that **approximation theory of tame functions matters** (Rev 6Gn6: "this paper as groundwork and a logical first step toward a general approximation theory for tame functions"), **the approximation-theoretic result extends the state of the art** (Rev 6Gn6: "This paper is a solid first step in the interesting direction of "deep learning of tame functions"), and that the code and supplementary material are useful for **reproducibility**.

We have incorporated changes into the revised manuscript, they appear in blue. We again thank all reviewers and the AC for their valuable time and effort.

Best regards and thanks,

Authors of Submission 4369

---

### Meta-Review · Area_Chair_5E1h · 2025-12-27

**Summary:**

The paper studies uniform approximation of compactly supported functions definable in o-minimal structures by piecewise polynomial functions. It claims the first theoretical bound on the approximation error and, leveraging recent piecewise linear approximation results of Boissonnat et al. (2023), proposes a mixed-integer programming formulation that encodes affine hyperplane trees describing piecewise polynomial approximations with polyhedral pieces.

Reviewers generally found the constructive aspect, namely, the procedure for computing a piecewise polynomial approximation, interesting. In terms of criticisms, they raised concerns about the scalability of the MIP formulation and, to varying degrees, viewed the experimental section as reflecting these inherent limitations. Beyond this shared concern, their criticisms focused on somewhat different aspects of the submission.

Reviewer 6Gn6 focused primarily on the approximation bound as the main contribution, seeking clearer clarification of its novelty and of the technical ingredients required to combine existing results. 6Gn6 was ultimately satisfied with the responses and explicitly indicated an intention to raise the score (from 4 to 6), suggesting strong overall sympathy to the paper’s direction. Reviewer W6es concentrated on the practical side, carefully examining the proposed MIP formulation and its claimed practical solvability. Their criticism centered on feasibility and scaling. Reviewer A4b4 focused mainly on fit and relevance, questioning the paper’s contribution and implications for DL (also indicating scalability as a crucial metric for practical relevance). They viewed the work as closer in spirit to theoretical approximation or optimization than to mainstream deep learning research.

Overall, the paper was viewed as lacking clear downstream consequences. As the theoretical component takes a strongly mathematical perspective centered on constructive approximation of definable functions, some reviewers questioned whether this emphasis aligns well with ICLR’s scope. On the practical side, although the proposed MIP formulation is well motivated, concerns were raised about its computational viability and the absence of concrete implications or new insights for deep learning, leading to further doubts about the paper’s overall fit for the venue.

**Reviewer Concerns:**

The issues identified in the summary paragraph were viewed as outstanding. In particular, the paper does not make a clear case for the scalability of the specific MIP formulation, a requirement that is especially important in DL contexts. While this concern is somewhat mitigated by the authors’ clarification that the focus is not on solvability of the MIP, the lack of concrete consequences or insights specific to deep learning remained unresolved (the work cited by the authors in response to A4b4 is oriented quite differently).

Some of the major issues that were largely resolved include the decision not to focus on practical solvability of the MIP within the paper, and the clarification that the class of definable functions considered is natural and easy to verify. In addition, remarks added later regarding the positioning of the work were also important in addressing several reviewer concerns.

**Reviewer Scores:**

Reviewer 6Gn6 was already willing to raise their score. Reviewer W6es appeared less likely to do so, given that their main concerns centered on feasibility and scalability. Reviewer A4b4 was unlikely to change their position, as the fit and relevance issues remained unchanged. Thus, the mismatch in scope and relevance seems unlikely to be resolved through further author–reviewer discussion.

---

### Decision · Program_Chairs · 2026-01-26

Reject